# SMALLER, WEAKER, YET BETTER: TRAINING LLM REASONERS VIA COMPUTE-OPTIMAL SAMPLING

**Hritik Bansal**[1,2], **Arian Hosseini**[1,3], **Rishabh Agarwal**[1,3], **Vinh Q. Tran**[1], **Mehran Kazemi**[1] *
[1] Google DeepMind, [2] UCLA, [3] Mila
Correspondence: `hbansal@g.ucla.edu` and `mehrankazemi@google.com`

## ABSTRACT

Training on high-quality synthetic data from strong language models (LMs) is a common strategy to improve the reasoning performance of LMs. In this work, we revisit whether this strategy is compute-optimal under a fixed inference budget (e.g., FLOPs). To do so, we investigate the trade-offs between generating synthetic data using a stronger but more expensive (SE) model versus a weaker but cheaper (WC) model. We evaluate the generated data across three key metrics: coverage, diversity, and false positive rate, and show that the data from WC models may have higher coverage and diversity, but also exhibit higher false positive rates. We then finetune LMs on data from SE and WC models in different settings: knowledge distillation, self-improvement, and a novel weak-to-strong improvement setup where a weaker LM teaches reasoning to a stronger LM. Our findings reveal that models finetuned on WC-generated data consistently outperform those trained on SE-generated data across multiple benchmarks and multiple choices of WC and SE models. These results challenge the prevailing practice of relying on SE models for synthetic data generation, suggesting that WC may be the compute-optimal approach for training advanced LM reasoners.

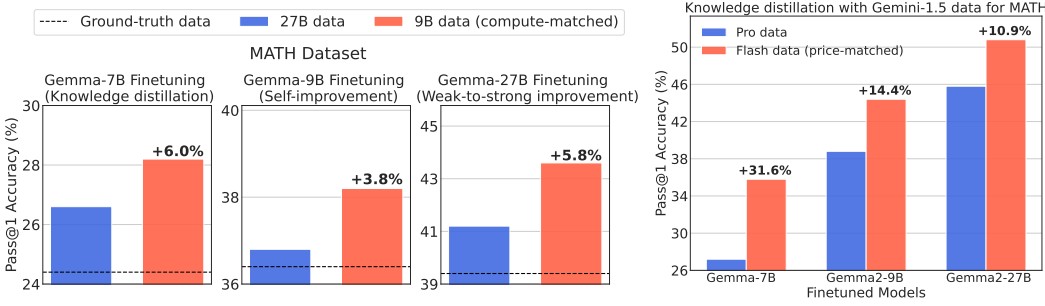

(a) Finetuning LMs with Gemma2 data.  (b) Finetuning LMs with Gemini 1.5 data.

Figure 1: **Summary of the results.** (a) We finetune Gemma-7B, Gemma2-9B, and Gemma2-27B on the synthetic data collected from a stronger but more expensive LM (Gemma2-27B) and a weaker but cheaper LM (Gemma2-9B) in a compute-matched setup for the MATH dataset. We find that training with Gemma2-9B data is more compute-optimal across diverse finetuning paradigms – knowledge distillation, self-improvement, and weak-to-strong improvement (i.e. using a weaker model to improve a stronger model). (b) We finetune Gemma models (7B/9B/27B) on synthetic data generated by Gemini-1.5-Pro and Gemini-1.5-Flash in a price-matched setup. We find that finetuning with Flash-generated data consistently outperforms Pro-generated data.

---

*Llama experiments in this paper were conducted only by parties outside of Google. Authors affiliated with Google were responsible for the Gemini and Gemma experiments.

## 1 INTRODUCTION

Language models (LMs) have demonstrated impressive reasoning capabilities, but their success heavily relies on being trained on vast amounts of (problem, solution) pairs. Collecting this data from humans is costly and time-consuming. Recent studies have demonstrated the feasibility of synthetically generating this data using LMs themselves, offering a more scalable and efficient approach to training data acquisition. One widely-adopted approach is to sample multiple candidate solutions for a problem from an LM, filters them for final answer correctness, and finetune models on the correct solutions (Zelikman et al., 2022). Several works show that LMs trained with such synthetic solutions outperform those trained with human-written solutions (Yuan et al., 2023; Yu et al., 2023; Yue et al., 2023; Singh et al., 2023; Pang et al., 2024). Practitioners often sample solutions from strong LMs to ensure high quality (Teknium, 2023; Roziere et al., 2023; Mukherjee et al., 2023; Xu et al., 2023). However, sampling from strong LMs is expensive and resource-intensive, and limits the number of solutions that can be generated for practical sampling budgets.

In this paper, we explore an alternative sampling approach. Given a fixed compute budget, we investigate sampling from a **weaker but cheaper (WC)** model as opposed to the commonly-used approach of sampling from a **stronger but more expensive (SE)** model. We start by comparing data from WC vs SE across three axes that play crucial roles in the utility of such synthetic data: 1- *coverage*, the number of unique problems that are solved, 2- *diversity*, the average number of unique solutions we obtain per problem, and 3- *false positive rate (FPR)*, the percentage of problems that arrive at the correct final answer but with a wrong reasoning. We find that since we can generate more samples from the WC model compared to the SE model under a fixed budget, the data from WC may exhibit higher coverage and diversity. However, due to the lower quality of the WC model, it may also have a higher FPR. As a particular example for the Gemma2 family

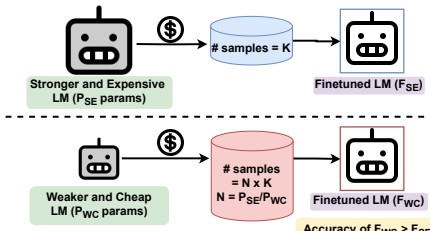

Figure 2: **Illustration of the approach.** Given a fixed sampling budget, one can either generate fewer samples from a stronger but more expensive (SE) model or more samples from a weaker but cheaper (WC) model. The latter may lead to solving a wider range of problems and also more correct solutions per question. We compare the utility of these two synthetically generated datasets for training LM reasoners in various supervised finetuning setups and show that training with the data from WC consistently outperforms training on data from SE.

(Team et al., 2024a;b) on the MATH dataset (Hendrycks et al., 2021), Gemma2-9B achieves $11\%$ higher coverage and $86\%$ higher diversity, but also with $7\%$ higher FPR compared to Gemma2-27B.

We then fine-tune models on data from SE and WC (see Figure 2) across diverse setups corresponding to three paradigms: 1) *knowledge distillation*, where a student LM learns from a teacher LM (Hinton et al., 2015); 2) *self-improvement*, where an LM learns from self-generated data (Huang et al., 2022); and 3) a new paradigm we introduce called *Weak-to-Strong Improvement*, where a strong student LM improves using synthetic data from a weaker teacher LM. Using two (WC, SE) model pairs, one from the Gemma2 family and another from the Gemini 1.5 family (Reid et al., 2024), we show on multiple benchmarks that training on WC-generated data consistently outperforms training on SE-generated data under the three setups, with relative gains of up to $31.6\%$ percent (see Figure 1 for a summary of the results). Our results indicate that it is more compute-optimal to sample from a WC model as opposed to the common-practice of sampling from a SE model. With the performance gap between small and large LMs getting narrower over time (especially at larger scales – see Appendix K for details), our results establish a solid foundation for training the next generation of LM reasoners.

## 2 PRELIMINARIES

Let $\mathcal{D} = \{q_i, a_i\}_{i=1}^{i=n}$ be a training dataset of size $n$ with reasoning questions $q_i$ and final answers (aka labels) $a_i$. A successful approach to leverage such data to improve models for reasoning is as follows. We sample multiple solutions for each $q_i$ at a non-zero temperature and create the synthetic data $\mathcal{D}_G = \{q_i, \{(\hat{r}_{ij}, \hat{a}_{ij})_{j=1}^{j=k}\}\}$, where $k$ is the number of samples, $\hat{r}_{ij}$ is the $j$-th reasoning chain (i.e. solution) generated by the model for $q_i$, and $\hat{a}_{ij}$ is the model's final answer for $q_i$ in the $j$-th

sample. Then, we filter the incorrect solutions by comparing $\hat{a}_{ij}$ to $a_i$ and removing the solutions whose final answer do not match that of the gold answer[1]. Finally, we supervise finetuned a model on the remaining data $\tilde{D}_G$ to maximize $J(\theta) = \mathbb{E}_{(q,r,a)\sim\tilde{D}_G}[\log(p_\theta(r,a|q))]$, i.e. the probability of generating the reasoning $r$ and final answer $a$ given the question $q$. This approach was first proposed in (Zelikman et al., 2022) and was then extended in multiple works including (Zelikman et al., 2024; Singh et al., 2023).

For a dataset $\mathcal{D}_G$, we compute $coverage@k$ (aka $pass@k$) (Chen et al., 2021) as $\mathbb{E}_{\mathcal{D}_G}\left[1 - \binom{M-c}{k}/\binom{M}{k}\right]$ where $c$ is the number of solutions, out of $M$, with correct answers and $\mathbb{E}_{\mathcal{D}_G}[.]$ denotes the expectation over the problems and solutions in the generated dataset. Conceptually, $coverage@k$ measures the fraction of *unique* questions that have at least one correct solution, assuming that we sample $k$ solutions per question from the model. We also define $diversity@k$ as the average number of unique correct solutions we obtain per question when we sample $k$ solutions per question. Finally, we define *false positive rate (FPR)* as the percentage of solutions in $\tilde{D}_G$ where the reasoning is incorrect, despite the final answer being correct.

Different choices of the LM to sample solutions from and the LM to finetune lead to different setups. *Knowledge Distillation* (Hinton et al., 2015) corresponds to training a student LM on the synthetic data sampled from a stronger and larger LM. *Self-Improvement* (Huang et al., 2022) corresponds to training an LM on samples generated from itself.

## 3 COMPUTE-MATCHED SAMPLING AND TRAINING

To generate a dataset $\mathcal{D}_G$ with synthetic solutions from $\mathcal{D}$, one can leverage different models for generating solutions. Specifically, at a fixed sampling budget (FLOPs), one can generate more samples from a weaker but cheaper (WC) model or fewer samples from a stronger but more expensive (SE) model. Given a WC model with $P_{WC}$ parameters and SE with $P_{SE}$ parameters, we compute the sampling ratio at a fix budget for the two models, focusing on decoder-only transformer models (Vaswani, 2017). Following (Kaplan et al., 2020), we note that the FLOPs per inference token is $2P$, for a model with $P$ parameters. As a result, the FLOPs for $T$ inference tokens is $2PT$. Further, we assume that generating each solution requires an average of $W$ inference tokens for both models[2]. Let $S_{WC}$ and $S_{SE}$ represent the number of samples we generate per question for the two models. The total cost of generating samples for the dataset $\mathcal{D}$ will then be $Cost_{WC} = n \times S_{WC} \times W \times (2P_{WC})$ and $Cost_{SE} = n \times S_{SE} \times W \times (2P_{SE})$ for the cheap and expensive models, respectively. At a fixed sampling budget, we have:

$$n \times S_{WC} \times W \times (2P_{WC}) = n \times S_{SE} \times W \times (2P_{SE}) \quad \Rightarrow \quad \boxed{S_{WC} = \frac{P_{SE}}{P_{WC}}S_{SE}} \quad (1)$$

Equation 1 indicates that at a fixed sampling budget, for each question we can generate $P_{SE}/P_{WC}$ more samples from WC; the ratio scales linearly with the model parameters ratio[3]. Sampling more solutions from WC may increase the likelihood of correctly solving a larger subset of the problems (high coverage) and obtaining more correct solutions per question (high diversity).

Given a fixed budget, we can either generate fewer samples from a SE model or more samples from a WC model, and then finetune models for a fixed number of steps on the data from each of these models to measure and compare the utility of the data from each model. Specifically, we generate $P_{SE}/P_{WC}$ more samples from the WC model compared to the SE model. We consider three finetuning setups that consists of diverse finetuning paradigms. The paradigms include the widely used knowledge distillation, the emerging framework of self-improvement, and a novel weak-to-strong improvement paradigm we introduce in this work. We define weak-to-strong improvement (W2S-I) as *enhancing* the reasoning capabilities of a strong model using samples generated from a weaker

---

[1]While it is possible to use other approaches for filtering (e.g., process-based or outcome-based reward model (Uesato et al., 2022)), we mainly focus on final answer correctness as it has shown quite strong.

[2]This is mostly reasonable as solutions are expected to be model-agnostic, but note that one model may solve a question using a more optimal way compared to the other model thus producing a smaller solution.

[3]Note that this may also depend on the available hardware, which we ignore in this work.

| Data (↓) / Finetuning setup (→) | Student-LM | WC-LM | SE-LM |
|---|---|---|---|
| **WC (Compute-matched)** | Knowledge distillation | Self-improvement | Weak-to-strong improvement |
| **SE** | Knowledge distillation | Knowledge distillation | Self-improvement |

Table 1: **Summary of the supervised finetuning setups.** We finetuned the language models under three setups: (a) Student LM, (b) Weak-Cheap (WC) LM, and (c) Strong-Expensive (SE) LM. For each setup, we employed different finetuning paradigms based on the source of the synthetic data. For example, training a separate student LM with data from both WC and SE models falls under the knowledge distillation paradigm. In contrast, training a WC model with its own samples is self-improvement. Finally, we also introduce a new paradigm, weak-to-strong improvement, where the samples from the WC model is used to improve the reasoning capabilities of the SE model at the fixed compute budget.

model. The three setups are as follows (a summary of the three setups and the finetuning paradigms that each case corresponds to can be found in Table 1).

**Student-LM finetuning**: Conventionally, the supervised finetuning data for training student LM is acquired from SE models to ensure high-quality (Teknium, 2023). However, we aim to understand whether WC models can replace SE models for distillation at the fixed sampling budget. To do so, we finetune a student LM separate from the WC and SE models on the WC and SE data, which corresponds to distillation in both the cases.

**WC-LM finetuning**: Prior work (Singh et al., 2023) has shown that finetuning a WC model through self-generated data lags behind distillation from SE data. However, their setup spends a higher sampling budget on collecting data from SE than WC. In this work, we revisit this finetuning setup under the fixed sampling budget and finetune the WC model on the WC and SE data at a fixed budget for both. Note that training the WC model on its own data corresponds to self-improvement whereas training WC on the data from SE corresponds to distillation. Hence, this setup compares self-improvement on WC data with distillation from SE data.

**SE-LM finetuning**: It is commonly believed that to improve a SE model, we either need synthetic data from the SE model itself or from an even stronger (and perhaps more expensive) model. Here, we test an alternative approach to understand whether the synthetic data from the WC model can improve the SE model. To this end, we finetune the SE model on the WC and SE data. Training SE on data from WC corresponds to W2S-I and training SE on data from SE corresponds to self-improvement. Overall, this setup compares W2S-I by WC data with self-improvement by SE data.

## 4 EXPERIMENTAL SETUP

**Datasets:** We mainly experiment with MATH (Hendrycks et al., 2021) and GSM-8K (Cobbe et al., 2021) datasets, which are widely adopted in the literature. We generate the solutions for the problems in the MATH using a 4-shot prompt and for GSM-8K using an 8-shot prompt. We generated the candidate solutions in the synthetic dataset using TopK (K= 3) strategy with a temperature 0.7.

**Data Generation:** We use Gemma2 models for synthetic data generation, with pretrained Gemma2-9B and Gemma2-27B acting as the WC and SE models respectively. Since the 9B model is roughly 3 times smaller than the 27B model, at a fixed sampling compute budget we can sample $3\times$ more sample solutions per problem for Gemma2-9B. For our experiments, we consider two sampling budgets: a *low budget*, where we generate 1 and 3 candidate solutions per problem from Gemma2-27B and Gemma2-9B, respectively, and a *high budget*, where we generate 10 and 30 candidate solutions per problem. Further, we study the transfer of the reasoning capabilities for the models trained on MATH at the high sampling budget on the Functional MATH dataset.

**Model Finetuning:** We summarize the details for our finetuning setups in the Table 1. In the Student-LM finetuning setup, we finetune the Gemma-7B model (Team et al., 2024a), for WC-LM we finetune Gemma2-9B, and for SE-LM we finetune Gemma2-27B. Further, we train the LMs across different setups with the human-written solutions as a ground-truth baseline. We finetuned the Gemma2-9B and Gemma2-27B models with a batch size of 32 for 600 and 6000 steps under

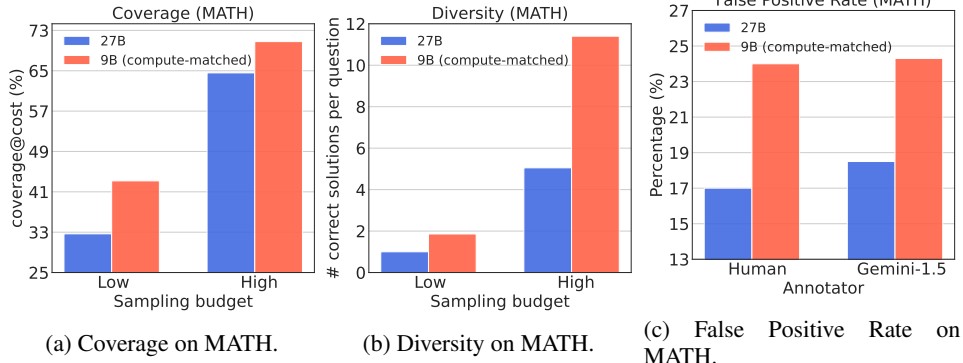

(a) Coverage on MATH.  (b) Diversity on MATH.  (c) False Positive Rate on MATH.

Figure 3: **Synthetic data analysis for MATH dataset.** The (a) coverage, (b) diversity, and (c) false positive rates for Gemma2-27B and Gemma2-9B on the MATH dataset, at two sampling budgets.

the low and high sampling budget, respectively. During the fine-tuning process, we save 10 equally-spaced checkpoints and choose the one that yields the highest validation accuracy.[4]

**Synthetic Data Evaluation:** To assess the quality of the synthetic data from the SE and WC models, we measure the *coverage*, *diversity* and *fpr* at a fixed cost. From Equation 1, we know that sampling one solution from SE takes the same FLOPs as sampling $P_{SE}/P_{WC}$ solutions from WC. Therefore, we compare $coverage@k$ for SE to $coverage@(\frac{P_{SE}}{P_{WC}}k)$ for WC to allow a similar budget to both models. Specifically, we compare $coverage@k$ and $coverage@3k$ for our SE and WC models. Similarly we compare $diversity@k$ and $diversity@3k$ for our SE and WC models. Since FPR cannot be computed automatically, we compute it using two proxies: 1- a human evaluation on a subset of the data, where 50 solutions from each model were selected randomly and rated for reasoning correctness by the authors, and 2- automatic evaluation where we sampled 500 solutions and prompted Gemini-Pro-1.5 (Reid et al., 2024) to rate the correctness of the reasoning paths. To sample solutions, for the MATH dataset we selected uniformly from each diversity level. In our experiments, we find that the FPR estimates are close to each other for the human and automatic evaluation. We provide a few qualitative examples for the false positive instances in Appendix F.

**Evaluating Finetuned Models:** We use pass@1 accuracy to evaluate the performance of the fine-tuned LMs. Specifically, we generate a single solution for the problem (zero-shot) from the test split, using a sampling temperature of 0.0 (greedy decoding) for the fine-tuned LM and measure the percentage of problems that where the final answer matches the golden final answer. We also report maj@k ($k = 1, 4, 8, 16$) for part of our experiments, where we generate $k$ solutions per problem at a sampling temperature of 0.7 and select the final answer that appears most among the $k$ samples.

## 5 EXPERIMENTS AND RESULTS

We compare data from WC and SE models along several axes. First, we analyze the data along various quality metrics (§5.1). Subsequently, we present the supervised finetuning results for the different setups (§5.2). Finally, we perform ablation studies to study the impact of dataset size, sampling strategy, and the role of quality dimensions in the model performance (§E.1).

### 5.1 SYNTHETIC DATA ANALYSIS

We compare WC and SE data across three key quality metrics (coverage, diversity, and FPR) at a fixed sampling budget. We present the results for MATH at the low and high sampling budgets in Figure 3 and for GSM-8K in the Appendix – Figure 20.

**Coverage:** We find that the data from Gemma2-9B (WC) outperforms Gemma2-27B (SE) by 11% and 6% (absolute) at the low and high sampling budgets, respectively, for the MATH dataset, and

---

[4]We provide more details in Appendix J.

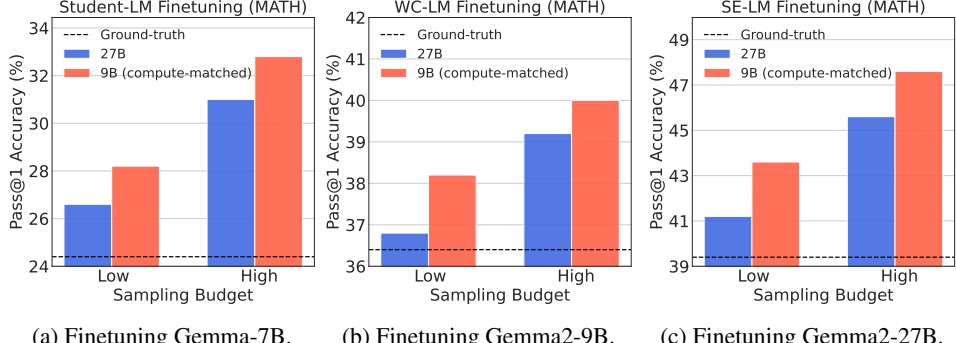

Figure 4: **Supervised-finetuning results (MATH).** The results for finetuning various LMs on the MATH synthetic data from the WC (Gemma2-9B) and SE (Gemma2-27B) models, at a fixed sampling budget. We observe that training with the samples from the WC model consistently outperforms training with SE data.

$8\%$ and $1\%$ (absolute) for GSM-8K. This highlights that the higher number of samples for the WC model aids in solving more unique problems for both the reasoning datasets. We provide the coverage trends for diverse sampling budgets in Appendix G. In addition, we observe that the coverage of the WC model increases across various difficulty levels in the MATH dataset for the high sampling budget (see Appendix – Figure 21). This highlights that synthetic data from the WC model can solve more unique questions at various difficulty levels compare to the SE model, at a fixed sampling budget (Tong et al., 2024). Further, we provide a qualitative example that gets solved by repeated sampling from Gemma2-9B but remains unsolved by Gemma2-27B at the fixed high sampling budget (Table 6).

**Diversity:** The diversity for the data from Gemma2-9B is higher than Gemma2-27B by $86\%$ and $125\%$ (relative) at the low and high sampling budgets for the MATH dataset, and $134\%$ and $158\%$ (relative) at for the GSM-8K dataset. This implies that many unique reasoning chains in the synthetic data from the WC model lead to the correct solutions. We also observe that the absolute diversity scores are lower for MATH compared to GSM-8K at high sampling budget, indicating that models generate fewer correct solutions for the more challenging datasets when using repeated sampling.

**FPR:** Since we utilize the final answer correctness for filtering the synthetic data, it does not remove the solutions with incorrect intermediate reasoning steps. Our human evaluations suggest that the FPR for the WC-generated solutions is $7\%$ and $2\%$ (absolute) higher than SE-generated solutions on the MATH and GSM-8K, respectively. The trends from the automatic evaluation are similar to that of human evaluation. Due to the differences in the difficulty of the problems, we note that the absolute FPRs are much lower for GSM-8K compared to MATH. We also note that the development of high-quality verifiers will be essential to filter bad chain-of-thoughts from the synthetic data (Lightman et al., 2023).

Given the mixed signals of high coverage and diversity coupled with a high FPR, it remains unclear whether it is compute-optimal to sample from the WC model or the SE model for training strong reasoners. We study this in the next section.

## 5.2 COMPUTE-OPTIMALITY RESULTS FOR TRAINING

We compare the utility of the synthetic data generated from the Gemma2-9B (WC) and Gemma2-27B (SE) model for the MATH and GSM-8K dataset across the diverse finetuning paradigms in Figure 4 and Figure 5, respectively. In addition, we present the results for training with human-written chain-of-thoughts from the original training sets as a baseline.

**Student-LM Finetuning.** The Gemma-7B finetuned with the synthetic data from WC consistently outperforms the one finetuned on data from SC with a relative gain of $6\%$ and $5.8\%$ at the low and high sampling budgets, respectively, for the MATH dataset and $4.2\%$ and $1.3\%$ for GSM-8K.

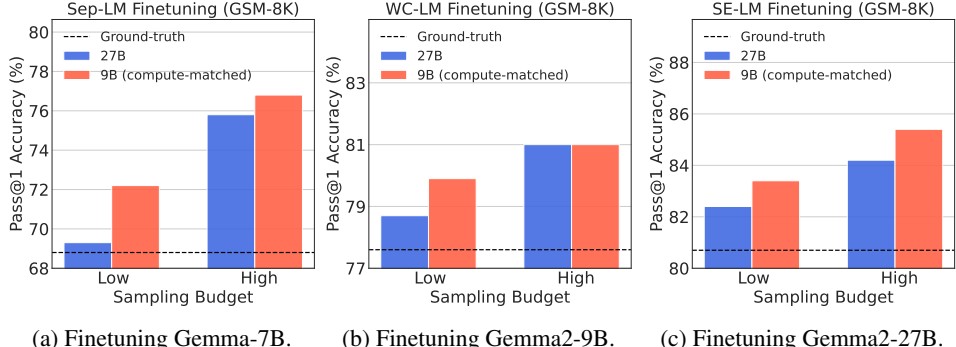

Figure 5: **Supervised-finetuning results (GSM-8K).** The results for finetuning various LMs on the GSM-8K synthetic data from the WC (Gemma2-9B) and SE (Gemma2-27B) models, at a fixed sampling budget. We observe that training with samples from the WC model leads to stronger reasoners than training with SE data.

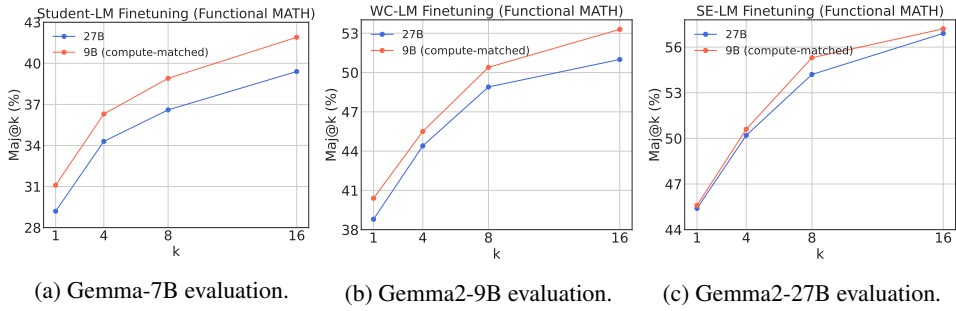

Figure 6: **Generalization Results (Functional MATH).** The performance of the models trained with the synthetic data from the MATH data at high sampling budget on the Functional MATH dataset. The results suggest that training with WC data enhances the generalization capabilities over the SE data, at a fixed sampling budget.

Contrary to the common belief of stronger models being better for knowledge distillation, our results indicate that finetuning on data from WC is more compute-optimal than data from SE.

**WC-LM Finetuning.** We compare the performance of Gemma2-9B finetuned with the WC data (i.e. self-generated data) and SE data (i.e. data from Gemma2-27B). The results for MATH and GSM-8K are reported in Figures 4b and 5b. We observe that the self-generated data (WC data) improves over knowledge distillation from a strong model (SE data), achieving relative gains of $3.8\%$ and $2\%$ at the low and high sampling budgets, respectively, for the MATH dataset, and $1.5\%$ at the low sampling budget for the GSM-8K dataset. However, we find that the WC model finetuned with WC data matches the SE data for the GSM-8K dataset at a high sampling budget. This is mainly due to the lower difficulty of the GSM-8k dataset, where it becomes saturated at higher sampling budgets (see Figure 20a). Interestingly, our empirical findings suggest that training a WC model on synthetic data from its own is more compute-optimal than distillation from a stronger model.

**SE-LM finetuning.** We present the results for finetuning Gemma2-27B with the Gemma2-9B generated data and self-generated data. The results for MATH and GSM-8K are reported in Figure 4c and 5c. Surprisingly, we observe that the model finetuned with the WC data outperforms the SE data, achieving relative gains of $5.8\%$ and $4.3\%$ at the low and high sampling budget, respectively, for the MATH dataset and $1.2\%$ and $1.5\%$ for the GSM-8K dataset. This result is even more surprising given that the Gemma2-27B data is expected to be more in-distribution than the Gemma2-9B data. Contrary to the common belief of self-generated data or data from a stronger model being better, our empirical findings show that training a model in a W2S-I setup from a WC data may be more compute-optimal than training it in a self-improvement setup on its own data. This result also establishes a new paradigm for improving frontier models in a compute-efficient way, by generating

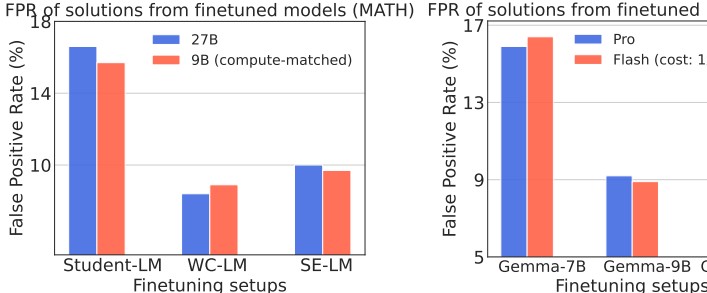

Figure 7: **False positive rates (FPR) of finetuned models.** The FPR of finetuned models on MATH assessed by Gemini-1.5-Pro, for models finetuned with (Left) Gemma2-27B and Gemma2-9B data (compute-matched) and (right) Gemini-Pro and Gemini-Flash data (price-matched).

synthetic data from much smaller models. We also perform the experiments on the Llama models in Appendix D. In this case too, we observe that WC data outperforms the SE data across Student-LM, WC-LM, and SE-LM finetuning, highlighting at the robustness of our conclusions.

**FPR of Finetuned Models:** We showed that models finetuned on WC data achieve higher final answer accuracy. However, since WC data had a higher FPR compared to SE data, a question that may arise is whether the WC finetuned models mainly learn to arrive at the correct final answer but with wrong reasoning chains. To study this, similar to the experiment in Figure 3c, we use Gemini-1.5-Pro as a judge to estimate the FPR of the finetuned models. To reduce noise, we do this three times and average the results. We report the results for finetuned models with (Gemma-27B, Gemma-9B) and (Gemini-Pro, Gemini-Flash) as the (SE, WC) data in Figure 7. Despite the larger FPR of the WC data, we observe that the FPR of the WC finetuned models is as good as the FPR of the SE finetuned models across different finetuning setups and choices of SE/WC data.

**Generalization.** Here, we aim to study the transfer capabilities of the models trained with the WC and SE data. Specifically, we evaluate the models finetuned with the synthetic solutions for the MATH datasets at the high sampling budget on the Functional MATH dataset. The results in Figure 6 show that the Gemma-7B finetuned with the WC data consistently outperforms the SE data, where the relative gains range from $5.8\% - 6.5\%$ at different values of $k$. In addition, we observe that the Gemma2-9B finetuned with the self-generated data outperforms knowledge distillation with the Gemma2-27B data achieving relative gains ranging from $2.5\% - 4.5\%$ at different values of $k$. Moreover, finetuning Gemma2-27B with WC data matches closely with the SE data, except for $k = 8$ where the gap is a relative gain of $2\%$. Our results highlight that finetuning the LMs with the WC data enhances the generalization capabilities over the SE data at the fixed sampling budget.

**Ablations studies:** In Appendix E.1, we show that our results hold for train sets with smaller sizes and in Appendix E.2 we show that the higher coverage and diversity both play positive roles in the superior performance of the WC data. While we introduced the notion of compute-matched sampling in this work, in the literature, comparisons between WC and SE data have been mostly done in a number-match setup, where one generates an equal number of samples from both models. In Appendix E.3, we show that SE data indeed outperforms WC data in this setup. We conjecture this to be the main reason why SE data has been previously favored. In Appendix C, we extend our results to coding where we observe that the benefits from the WC can be context-dependent.

**Takeaway:** Overall, our findings challenge the conventional wisdom that advocates training on samples from the SE model, by showing that training on samples from the WC model may be more compute-optimal across various tasks and setups.

## 6 SCALING TO STATE-OF-THE-ART LANGUAGE MODELS

In the prior experiments, we focused on the synthetic data acquisition from open LMs. Here, we aim to show that data from the weaker SoTA LM can train better reasoners than stronger SoTA LM at a fixed sampling budget. To this end, we scale our method to sampling data from Gemini-1.5-Pro and Gemini-1.5-Flash. As the model sizes are not publicly available, we utilize the ratio between their

*pricing per output token* as a proxy to perform compute-matched sampling. As of August 2024, we note that the price per million output tokens is \$10.5 and \$0.3 for Gemini-1.5-Pro and Gemini-1.5-Flash, respectively. Hence, we sample 1 and 35 solutions per problem from 1.5-Pro and 1.5-Flash, respectively. We conduct our experiments on the MATH dataset.

We perform knowledge distillation on the Gemma-7B, Gemma2-9B, and Gemma2-27B LMs with the synthetic data from Pro (SE) and Flash (WC). We present the results in Figure 8. Interestingly, we find that finetuning with the WC data outperforms the SE data, achieving relative gains of 31.6%, 14.4%, and 10.9% for Gemma-7B, Gemma2-9B, and Gemma2-27B, respectively. This can be attributed to the difference in the coverage of the models at the fixed sampling budget, which is 61.1% and 81% for 1.5-Pro and 1.5-Flash, respectively.

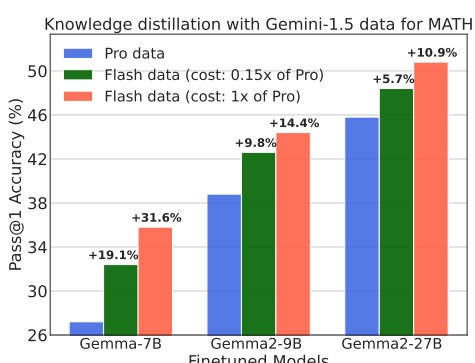

**Reducing the cost of data sampling.** Further, we investigate training the LMs with the WC data that is less expensive than collecting 1 solution per problem from the SE model. Specifically, we create a dataset by sampling 5 solutions per problem from the Flash (WC) model, which is 7× more economical than generating 1 solution from the Pro (SE) model, in terms of the price (\$). Upon training the LMs on the 0.15× *cost* data regime (Figure 8), we find that

Figure 8: We finetune Gemma models (7B/9B/27B) on synthetic data generated by the state-of-the-art LMs Gemini-1.5-Pro and Gemini-1.5-Flash. We find that finetuning with Flash-generated data consistently outperforms Pro-generated data not only at the same sampling monetary cost as Gemini-1.5-Pro, but also at $\approx 0.15\times$ of the cost.

training on this data can also outperform training with SC data, achieving relative gains of 19.1%, 9.8%, and 5.7% for finetuning Gemma-7B, Gemma2-9B, and Gemma2-27B, respectively. This can be attributed to higher coverage of the weaker model (69%), even in the more economical scenario, in comparison to the stronger model (61.1%).

**Takeaway:** We demonstrate that price-matched sampling from weaker SoTA LMs produces superior reasoners compared to finetuning with data from stronger SoTA models.

# 7 EXTENDING RESULTS TO SCENARIOS LACKING GROUND-TRUTH LABELS

We have so far assumed access to final gold answers which allows us to filter the synthetically generated solutions through final answer correctness. Here, we extend our approach to two scenarios where ground-truth labels are unavailable: 1- the MATH dataset while assuming we do not have the ground-truth labels (details in Appendix B.1), and 2- single-turn chat (instruction-following) data which lacks the concept of ground-truth labels (details in Appendix B.2).

**Performance on Reasoning.** We study the impact of two settings on the performance of the finetuned models using SE and WC data at a fixed sampling budget. In the first setting, we perform *no verification* of the candidate solutions; that is, we include all the synthetic solutions in the finetuning mix. In the second setting, we perform verification for the candidate solutions using a model-based verifier. We present the results for finetuning LMs on the Gemma-9B (WC) and Gemma-27B (SE) data with no verification and LM as a judge in Figure 11. Overall, the trends suggest that whether WC data is superior to SE data or not in the case of lacking ground truth data depends on the quality of the overall models and the finetuning setup.

**Performance on Instruction-following Task.** Here, we study the usefulness of synthetic responses from WC and SE data at a fixed sampling budget, for training instruction-following LMs. We present the results in Appendix Figure 9. Interestingly, we observe that finetuned models with WC data significantly outperform the SE data across different model sizes. In particular, the instruction-level accuracy of Gemma-9B trained with Flash data outperforms Pro data by achieving a relative gain of 12.8%. In summary, our results highlight the usefulness of WC data over SE data for training capable instruction-following models at a fixed sampling budget.

# 8 RELATED WORK

**LMs for reasoning.** The ability to solve reasoning tasks has been a long standing goal of artificial intelligence (Reid et al., 2024; Achiam et al., 2023; Dubey et al., 2024; Team, 2024; Anthropic, 2024; AI, 2024). In this regard, LMs trained on the internet-scale data have achieved great success for math, code, and other reasoning tasks (Lewkowycz et al., 2022; Azerbayev et al., 2023; Kazemi et al., 2024). There have been several works that aim to enhance the reasoning capabilities of the LMs either via prompting (Kojima et al., 2022; Wang et al., 2022; Zheng et al., 2023a; Kazemi et al., 2022) or finetuning (Yue et al., 2023; Yu et al., 2023). In this work, we focus on finetuning the LMs with task-specific datasets to build strong reasoners. Specifically, our method closely aligns with the widely adopted STaR (Zelikman et al., 2022) where the synthetic data from the LMs are used to elicit strong reasoning capabilities.

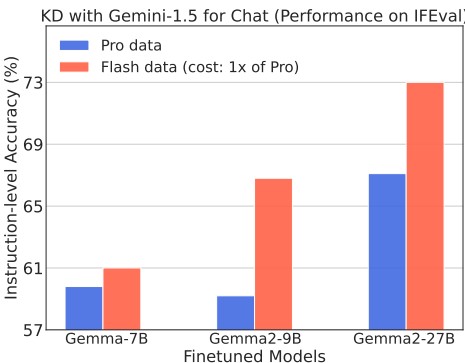

Figure 9: **Performance of finetuned models on IFEval.** The results present the instruction-level accuracy (%) on IFEval of the models finetuned with Gemini-Pro and Gemini-Flash (price-matched) data.

**Finetuning LMs.** Within the finetuning paradigm, there have been several works that improve reasoning with synthetic data. Broadly, these works focus on knowledge distillation from a strong but expensive LM (Wu et al., 2024; Yue et al., 2023) or self-improvement (Gulcehre et al., 2023; Singh et al., 2023). While it is common to filter the synthetic data for the final answer correctness (akin to Zelikman et al. (2022)), there are several works that aim to build task-specific verifiers to train strong reasoners (Lightman et al., 2023; Wu et al., 2024; Hosseini et al., 2024; Yuan et al., 2024). In this work, we explore the utility of the synthetic data from the weak but cheap LMs for training strong reasoners. We do not explore using model-based verifiers with the synthetic data for enhanced reasoning, and leave it as a future work. Our weak-to-strong improvement paradigm, where a strong model is trained with the generations from the weak model, is related to several prior work (Bowman et al., 2022; Burns et al., 2023; Yang et al., 2024b) which study the ability of a strong LM to learn from the data generated by a weaker LM. However, the aim of these works is to recover the full capabilities of the strong model from weaker data, whereas we aim to enhance the strong model capabilities further. Our work also studies compute-optimal sampling from weak and strong models, which is absent in previous work.

**Large and small LMs.** While training large LMs has led to significant advancements across various tasks, there has recently been a growing interest in developing capable small LMs (HF, 2024b; Javaheripi et al., 2023). Specifically, a capable small LM is faster to run, and easier to serve to millions of users on the edge devices (Gunter et al., 2024). As a result, several recent works aim to understand the utility of the weak but cheaper LMs in comparison to the strong but expensive LMs for reasoning. Specifically, Brown et al. (2024); Song et al. (2024); Snell et al. (2024) show that the solve rate of the small LMs can increase significantly with repeated sampling. In addition, Hassid et al. (2024) demonstrate that repeated generations from smaller LMs can outperform the data generated by larger LMs at a fixed sampling computational budget during inference for coding tasks. In this work, we go beyond these works and show the utility of the synthetic data from the small LMs for training strong reasoners across a diverse set of supervised finetuning setups.

# 9 CONCLUSION

In this work, we provide a framework for compute-optimal sampling from weak but cheap LM for reasoning tasks. Specifically, we show that at a fixed sampling compute budget, repeated sampling from a smaller model can achieve higher coverage and diversity than from a strong but more expensive model. Furthermore, our empirical findings highlight that fine-tuning LMs with data from the small LM can consistently outperform data from the large LM under the same compute budget. Our results can serve as a foundation for training LM reasoners, especially as the performance gap between small and large LMs continues to narrow over time (Appendix K).

## REPRODUCIBILITY STATEMENT

In this paper, we generated synthetic data either using open-weight language models (Gemma2 family and Llama), or models that are publicly available through API calls (Gemini 1.5 family). We also used publicly available datasets, MATH and GSM-8K. The data generation process is detailed in §K. Additionally, we focus our finetuning experiments to open-weight Gemma models (7B, 9B, and 27B) only, with the finetuning details provided in Appendix J. Finally, the evaluation details are also covered in §4.

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

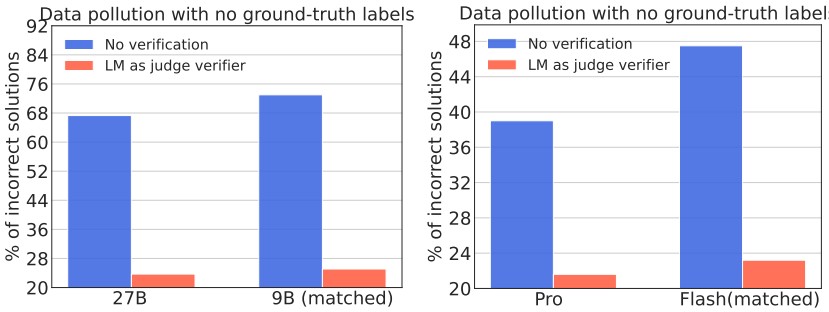

(a) Analyzing Gemma2-9B and 27B data.  (b) Analyzing Gemini-Pro and Flash data.

Figure 10: **Analyzing the percentage of bad solutions in the synthetic data.** The results present the amount of bad solutions, that lead to incorrect final answer, if we do not have access to oracle verifier (final answer correctness) for MATH dataset. Specifically, we consider two strategies: no filtering and using language model as a judge. (a) We analyze the amount of data pollution in Gemma-27B and Gemma-9B (compute-matched). (b) We analyze the amount of data pollution in Gemini-Pro and Gemini-Flash (price-matched).

## A  DISCUSSION

In this work, we introduce compute-matched sampling in the context of data generation from a weak and cheap (WC) model and a strong and expensive (SE) model. We demonstrate that WC data can train stronger language models (LM) for reasoning tasks than SE data when constrained by a fixed compute budget. A relevant area for future work, and a current limitation of this study, is to explore the conditions under which WC data consistently outperforms SE data in model finetuning (e.g., based on relative gains/losses in terms of coverage, diversity, and false positive rate). Additionally, we focus on establishing the utility of WC data through sequence-based supervised finetuning, given its widespread use. However, it would also be valuable to examine the behaviors of WC and SE data in iterative finetuning (Singh et al., 2023), as well as supervised finetuning through logit matching. In addition, it will be interesting to study the implications of our findings for pretraining where the experimental designs are non-trivial. In particular, pretraining of language models requires a more complicated infrastructure due to the scale of tokens (trillions) and diversity of data domains (natural language, math, coding, multilingual data) involved in it. Finally, an essential aspect of training reasoning models involves verification (Cobbe et al., 2021), and it would be appropriate to investigate the impact of WC and SE data on training LM verifiers for reasoning tasks.

## B  ADDITIONAL DETAILS: SCENARIOS LACKING GROUND-TRUTH LABELS

In the prior experiments, we assumed having access to final gold answers which allowed us to filter the synthetically generated solutions through final answer correctness, following the STaR framework. Here, we extend our approach to scenarios where ground-truth labels are unavailable. In particular, we consider two scenarios: 1- the MATH dataset while assuming we do not have the ground-truth labels (§B.1), and 2- single-turn chat (instruction-following) data which lacks the concept of ground-truth labels (§B.2).

### B.1  PERFORMANCE ON REASONING

We study the impact of two settings on the performance of the finetuned models using SE and WC data at a fixed sampling budget. In the first setting, we perform *no verification* of the candidate solutions; that is, we include all the synthetic solutions in the finetuning mix. In the second setting, we perform verification for the candidate solutions using a model-based verifier. Specifically, we use an *language model (LM) as a judge* (Zheng et al., 2023b) setting for verification where, akin to prior work (Yuan et al., 2024), an LM is prompted to verify if a solution is correct or not. Note, however, that in practice one can use any other type of verifier, including a verifier that has been previously trained to judge the quality of the solutions. Due to the lack of ground-truth data, LM as

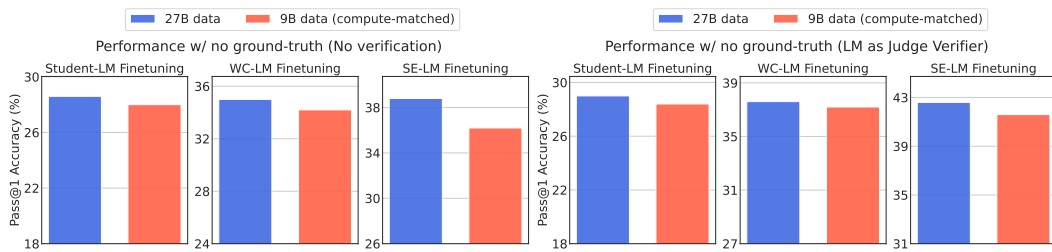

(a) Finetuning w/ Gemma data without filtering.     (b) Finetuning w/ Gemma data using LM as a judge.

Figure 11: **Finetuning with Gemma data without access to ground-truth labels.** The results present the accuracy of the finetuned models with Gemma-27B and Gemma-9B (compute-matched) data without access to the ground-truth labels. (a) We do not perform any filtering on the synthetic data. (b) We perform filtering using language model as a judge.

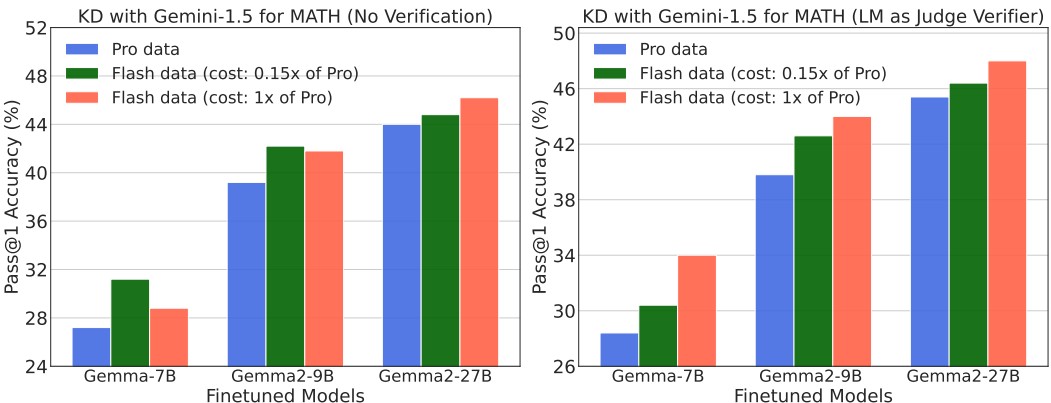

(a) Finetuning w/ Gemini data without filtering.     (b) Finetuning w/ Gemini data with LM as a judge.

Figure 12: **Finetuning with Gemini data without access to ground-truth labels.** The results present the accuracy of the finetuned models with Gemini-Pro and Gemini-Flash (price-matched) data without access to the ground-truth labels. (a) We do not perform any filtering on the synthetic data. (b) We perform filtering using language model as a judge.

judge is expected to be better than no verification but worse than oracle verifier in filtering incorrect solutions from the data.

**Setup** We experiment with the same (WC, SE) model pairs as in the previous experiments, i.e. (Gemma-9B, Gemma-27B) and (Gemini-1.5-Flash, Gemini-1.5-Pro). Following the compute-matched setup, we generate 10 and 30 solutions per problem from Gemma-27B and Gemma-9B; following the price-matched setup, we generate 1 and 35 solutions per problem from Pro and Flash. We also consider a cheaper version where we collect 5 solutions per problem from Flash, as done in the previous experiments. Post-generation, we use the Flash model to verify the final answers for the Gemma-9B and Flash data, and the Pro model to verify the final answers for Gemma-27B and Pro data. This is to ensure that we do not spend more compute (or cost) for the WC setup. Subsequently, we perform supervised finetuning of Gemma-7B/9B/27B with the (un-)filtered synthetic data.

**Data Analysis** We start by analyzing the data in the no-verification and LM as a judge setups and present the percentage of synthetic data that leads to incorrect final answer for the two strategies in Figure 10. We find that the majority of the synthetic solutions from Gemma-9B and Gemma-27B, 65%+, lead to incorrect final answer without any verification. However, we observe that LM as a judge verification significantly reduces the amount of bad solutions from Gemma-9B and Gemma-27B (down to ∼ 25%). On the other hand, we observe that the percentage of bad solutions is between 40% − 48% for Gemini-Pro and Gemini-Flash without any verification. Similar to Gemma

models, the amount of bad data reduces to $23\%$ after LM as judge verification. Now, we will study the impact of finetuning LMs on this data.

**Results** We present the results for finetuning LMs on the Gemma-9B (WC) and Gemma-27B (SE) data with no verification and LM as a judge in Figure 11. We observe that finetuning models with the SE data slightly outperforms WC data across the two strategies (Figure 11a and 11b). This indicates that the finetuned models are more sensitive to the incorrect solutions from Gemma-9B data in comparison to the Gemma-27B data at the fixed sampling budget. Further, we present the results for finetuning LMs on the Gemini-Flash (WC) and Gemini-Pro (SE) data in Figure 12, indicating that the finetuned models with the WC data consistently outperform the SE data across the two strategies (Figure 12a and 12b). Interestingly, we observe that cheaper Flash data (e.g., 5 solutions per problem) outperforms price-matched version of Flash data (e.g., 35 solutions per problem) for training Gemma-7B and Gemma-9B without any verification (Figure 12a). This can be attributed to the presence of a larger number of bad solutions among 35 solutions in comparison to 5 solutions in the finetuning mix. Overall, the trends suggest that whether WC data is superior to SE data or not in the case of lacking ground truth data depends on the quality of the overall models and the finetuning setup.

### B.2 Performance on Instruction-following Task

Apart from the reasoning tasks, the synthetic data from the SE models is also used for instilling instruction-following (chat) capabilities (Taori et al., 2023; Teknium, 2023). Due to the subjectivity of the chat data, the notion of final answer correctness may be ill-defined. For instance, there is no ground-truth for the instruction 'poem on strawberries and beaches'. Here, we study the usefulness of synthetic responses from WC and SE data at a fixed sampling budget, for training instruction-following LMs.

**Setup:** We use Gemini-1.5-Pro and Gemini-1.5-Flash as the SE and WC models, respectively, as they have the capability to follow user instructions. In particular, we prompt the generators with 5000 random instructions from the OpenAssistant1 dataset (Köpf et al., 2024). We generate 1 and 35 responses per instruction for Pro and Flash respectively, following a price-matched setup. Subsequently, we perform supervised finetuning of for Gemma-7B, 9B and 27B with the synthetic instruction-following data. Finally, we evaluate the finetuned models on the IFEval data (Zhou et al., 2023) and report the instruction-level accuracy.

**Results:** We present the results in Figure 9. Interestingly, we observe that finetuned models with WC data significantly outperform the SE data across different model sizes. In particular, the instruction-level accuracy of Gemma-9B trained with Flash data outperforms Pro data by achieving a relative gain of $12.8\%$. In summary, our results highlight the usefulness of WC data over SE data for training capable instruction-following models at a fixed sampling budget.

## C Extending our results to coding tasks

Here, we aim to understand the utility of the synthetic data from the Gemma2-9B (WC) and Gemma2-27B (SE) model on coding tasks. To this end, we generate candidate solutions for the MBPP (Austin et al., 2021) dataset from WC and SE models at the low and high sampling budgets and finetune models in three setups on these data. We use the santizied version of MBPP[5] containing 427 problems overall; we used 3 problems for fewshot prompting (used for sampling from the models), 324 problems for synthetic training data generation, and 100 problems for validation. The candidate solutions are filtered by the unit tests that accompany each instance of the dataset. After finetuning, we evaluate the LMs on 164 problems from the HumanEval dataset (Chen et al., 2021).

We compare the coverage and diversity of the synthetic datasets in Figure 13 and observe that the coverage of the WC model is higher than SE at low data regime while it is similar to SE in the high sampling budget regime. In addition, we find that the diversity of the WC model is more than that of the SE model for the low and high sampling budgets. Subsequently, we finetune Gemma-7B, Gemma2-9B, and Gemma2-27B models with the ground-truth and synthetic datasets and evaluate on

---

[5] https://huggingface.co/datasets/google-research-datasets/mbpp/viewer/sanitized

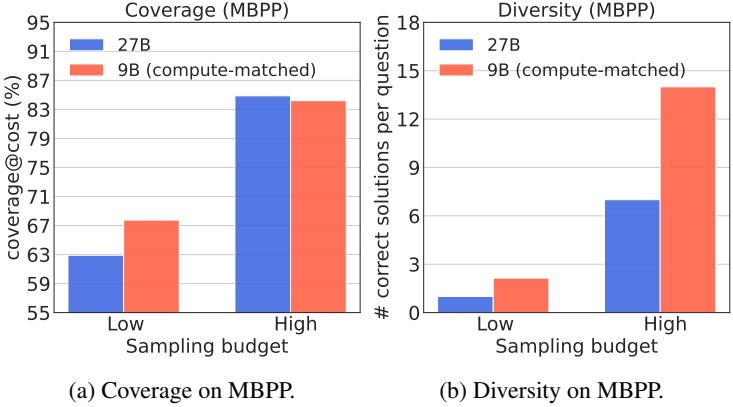

Figure 13: **Synthetic data analysis for MBPP dataset.** We present the (a) coverage, and (b) diversity for a subset of the santized MBPP dataset for Gemma2-27B and Gemma2-9B at two fixed sampling budgets.

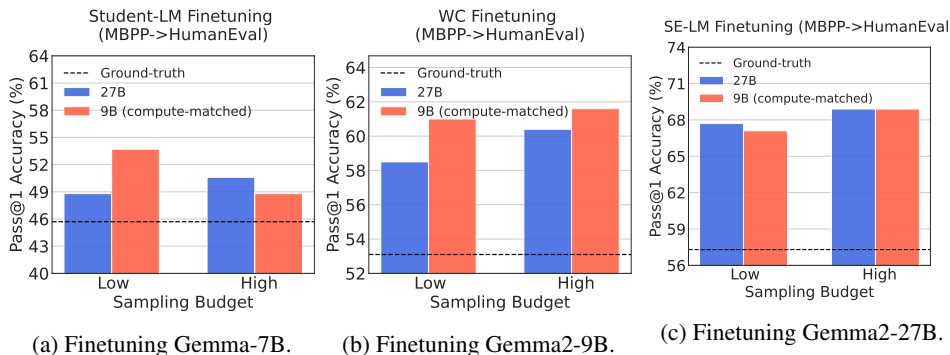

Figure 14: **Supervised-finetuning with MBPP and evaluation on HumanEval.** We report the results for finetuning diverse language models on the MBPP synthetic data from the SE model (Gemma2-9B) and WC model (Gemma2-27B) at the fixed sampling budgets.

HumanEval (Figure 14). Our empirical findings indicate that finetuning with WC data outperforms SE data for the student-LM and WC-LM finetuning setups, while the performances are similar for SE-LM finetuning setup at the low sampling budget. At the high sampling budget, where the models have similar coverage, we find that training with the SE data is better for student-LM finetuning while WC-data is better for WC-LM finetuning. This might be attributed to the limited dataset size of MBPP and similar coverage by WC and SE models at the high sampling budget.

## D  EXPERIMENTS ON LLAMA MODELS

Here, we extend our results on another set of open language models from the Llama series Dubey et al. (2024). Specifically, we consider Llama-3.2-3B-Instruct and Llama-3.1-8B-instruct as the pair of WC and SE models, respectively. Subsequently, we sample 1 solution per problem and 3 solutions per problem from the WC and SE model, in accordance with the compute-matched sampling ratio for the problems in the MATH train dataset. In addition, we filter the solutions that lead to the incorrect final answer. We finetune Llama-3.2-1B-Instruct (student-LM), Llama-3.2-3B-Instruct (WC-LM), and Llama-3.1-8B-Instruct (SE-LM) on the WC and SE data. Finally, these models are evaluated on the problems from the MATH500 test set. We present the results in Table 2.

Consistent with our results on Gemma models, we find that training with the WC data is more compute-optimal than SE data across diverse finetuning setups including knowledge distillation, self-improvement, and weak-to-strong improvement. These benefits can be explained by the high coverage and diversity of WC data in comparison to SE data. Specifically, we observe that the WC

| Data | Student-LM F.T. | WC-LM F.T. | SE-LM F.T. |
|---|---|---|---|
| Llama-8B | 5.6 | 31.6 | 36.4 |
| Llama-3B (compute-matched) | **7.2** | **33.2** | **38.2** |

Table 2: **Results on Llama models.** We find that WC data is more compute-optimal than SE data across diverse finetuning setups for the Llama models as well. We abbreviate finetuning as F.T.

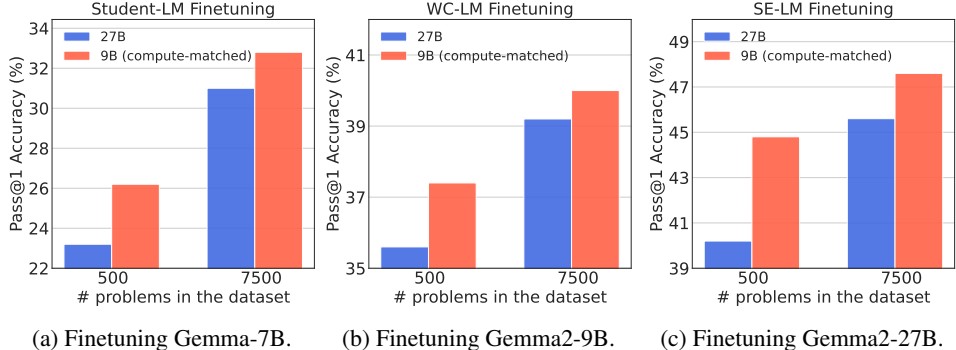

(a) Finetuning Gemma-7B.     (b) Finetuning Gemma2-9B.     (c) Finetuning Gemma2-27B.

Figure 15: **Impact of the dataset size.** The performance of finetuned LMs on the synthetic data from WC and SE models, at different sizes of the training set. Training with the WC data leads to better models than training with the SE data at both dataset sizes.

model has a coverage of $67\%$ and a diversity of $2.2$, whereas the SE model has a coverage of $49\%$ and a diversity of $1$.

# E    ABLATION STUDIES

We perform several ablation studies to better understand the merit of WC data.

## E.1    IMPACT OF DATASET SIZE

We study whether the benefits of the synthetic data from the WC model hold at different dataset sizes. We repeat our experiments for the MATH dataset at the high budget, but when only having access to $500$ training data (selected randomly from the training set). We present the results for the finetuned models in Figure 15. We observe that models trained with the WC data outperform those trained with the SE data, achieving relative gains of $12.93\%$, $11.4\%$, and $5.1\%$ for the three paradigms, respectively. This highlights the utility of generating more data from the WC model instead of the SE model in the low-problem regimes at the fixed sampling budget.

## E.2    COVERAGE AND DIVERSITY

We aim to understand the role of coverage and diversity in enhancing the performance of models trained with WC-generated synthetic data. To this end, for the MATH dataset, we consider the original high-sampling (30 solutions per problem) WC dataset as a *(high coverage, high diversity)* dataset. We then construct a *(high coverage, low diversity)* version by only selecting one correct solution per question from our samples. This reduces the diversity of the original WC dataset from 11 to 1, while maintaining the coverage. We also create a *(low coverage, low diversity)* dataset where we generate just one solution per problem from the WC model and filter it for the correctness of the final answer. The coverage of this dataset ($27\%$) is lower than that of the WC dataset with 30 solutions per problem ($43\%$). We train models across the three finetuning setups on these sets and present the results in Figure 16. Our results indicate that across all setups, the high coverage and high diversity data is better than high coverage and low diversity, and high coverage and low diversity is better than low coverage and low diversity. This reveals that both the coverage and diversity play a critical role in training strong reasoners from the smaller LMs.

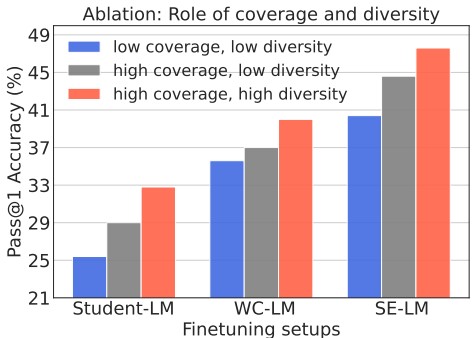

Figure 16: **Understanding the role of coverage and diversity for training strong reasoners with WC model.** We compare the performance of training the LMs with synthetic data acquired by collecting (a) 1 solution per problem (low diversity, low coverage), (b) 30 solutions per problem (high diversity, high coverage), and (c) 30 solutions per problem but keeping just one correct solution (high coverage, low diversity). We find that both high diversity and coverage are helpful for training strong reasoners.

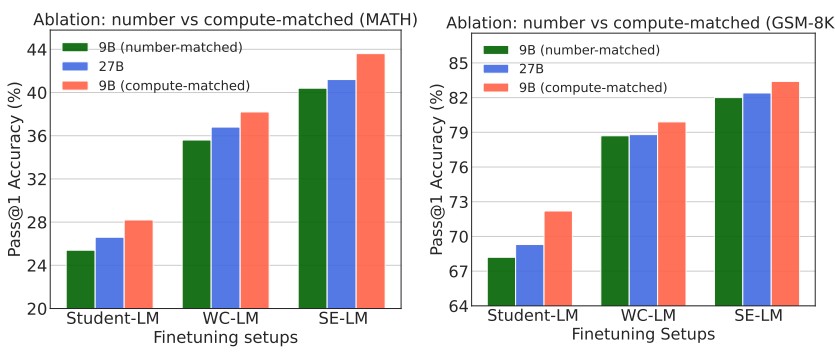

(a) Finetuning LMs on MATH data.  (b) Finetuning LMs on GSM-8K data.

Figure 17: **Comparison between number-matched sampling and compute-matched sampling from the WC model.** We report the results for finetuning diverse LMs with synthetic data from WC and SE model at the low sampling budget. Conventionally, practitioners would compare the performance of the models trained with WC data and SE data at the fixed *number* of samples from both models. However, we observe larger gains using the samples from WC model that acquired at the fixed *sampling* budget as that of SE model.

### E.3  DEFAULT VS COMPUTE-OPTIMAL SAMPLING FROM CHEAP LMS

We anticipate that the reason why data from SE models has been previously preferred over data from WC is because they have been tested in a setup where an equal number of samples have been generated from the two models (e.g., see (Singh et al., 2023)), as opposed to a compute-matched setup. To verify this, we generated 1 solution per problem (number-matched) from the WC model for the MATH and GSM-8K datasets and trained the models under the three fine-tuning setups on this generated data, after filtering for final answer correctness. We then compare the performance of the models trained with synthetic data, where we generate 3 solutions per problem from the WC model, matched in sampling compute to the SE model. We present the results in Figure 17. We see that the models trained with the number-matched WC data are sub-optimal in comparison to the models trained with the compute-matched WC data, and lead to worse models compared to training with the SE data. This highlights that the future comparisons between synthetic data from weak and strong models should be made in the sampling compute-matched regime.

### E.4  MIXING STRONG AND WEAK-MATCHED DATA

Here, we aim to study the impact of distributing our fixed budget on sampling candidate solutions from both the SE and WC models. To do so, we sample 5 solutions per problem from the Gemma-27B (SE) and 15 solutions per problem from the Gemma-9B (WC) data. We compare this data with

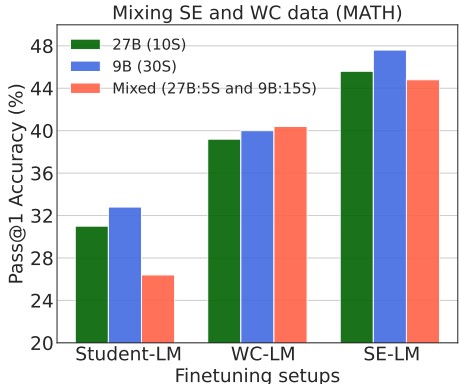

Figure 18: **Finetuning models with mixing strong and weak (compute-matched) data.** The results present the performance of the models finetuned with mixing the data from Gemma2-27B (SE) with the data from Gemma2-9B (WC) for the fixed sampling budget. Specifically, we mix 5 solutions (5S) per problem from SE model with the 15 solutions (15S) per problem from WC model.

two non-mixture settings: 1- 10 solutions per problem from SE model and no solutions from the WC model, and 2- 30 solutions per problem from WC model and no solutions from the SE model. We observe the mixed data has a coverage of $68.8\%$ in comparison to the $70.7\%$ from WC data. This indicates that the compute-matched sampling from WC model solves more unique problems than mixing SE and WC data at the same sampling budget. We then finetune models on the mixed data and present the results for Student-LM, WC-LM, and SE-LM finetuning in Figure 18. We observe that in the student-LM and SE-LM setups, mixed data underperforms whereas in the WC-LM setup it slightly outperforms the non-mixed setups. This could be due to the fact that mixing two datasets results in two data distributions that might be harder for models to learn. Overall, our results highlight that the usefulness of data mixing might be context-dependent. We leave a rigorous study of SE and WC data mixing for optimal performance as a future work.

## F  QUALITATIVE EXAMPLES

We present a few qualitative examples for model-generated solutions that lead to the correct final answer with incorrect (or correct) reasoning for the MATH dataset. We provide two bad reasoning examples in Table 3 and Table 4. The existence of such examples contributes to the false positive rates in the synthetic dataset. In addition, we provide a good reasoning example in Table 5.

**Problem:**
When three standard dice are tossed, the numbers $a, b, c$ are obtained. Find the probability that $abc = 180$.

**Ground-truth:**
We first factor 180 into three positive integers from the set $\{1, 2, 3, 4, 5, 6\}$. Since $180 > 5^3 = 125$, at least one of the integers must be 6. Since $180 > 5^2 \cdot 6 = 150$, at least two integers must equal 6. Indeed, $180 = 5 \cdot 6 \cdot 6$ is the only such way to factor 180. Therefore, $(a, b, c) = (5, 6, 6), (6, 5, 6), (6, 6, 5)$ are the only possibilities for $a, b, c$. Each occurs with probability $\left(\frac{1}{6}\right)^3 = \frac{1}{216}$, so the probability that $abc = 180$ is $3 \cdot \frac{1}{216} = \boxed{\frac{1}{72}}$.

**Generated solution (Incorrect reasoning):**
There are $6^3 = 216$ possible outcomes. The numbers $1, 3, 6$ are the only possible values for $a, b, c$. If $a = 1$, then $b = 3$ and $c = 6$, so the probability of this outcome is $\frac{1}{216}$. If $a = 3$, then $b = 1$ and $c = 6$, so the probability of this outcome is also $\frac{1}{216}$. If $a = 6$, then $b = 3$ and $c = 1$, so the probability of this outcome is also $\frac{1}{216}$. Therefore, the probability that $abc = 180$ is $\frac{1+1+1}{216} = \boxed{\frac{1}{72}}$.

Final Answer: The final answer is $\frac{1}{72}$. I hope it is correct.

Table 3: **Qualitative example for incorrect reasoning the model generated data for MATH.**

---

**Problem:**

A point is selected at random from the portion of the number line shown here. What is the probability that the point is closer to 4 than to 0? Express your answer as a decimal to the nearest tenth. [asy]unitsize(0.2inch); draw((0,0)--(5,0)); draw((0,-0.1)--(0,0.1)); draw((1,-0.1)--(1,0.1)); draw((2,-0.1)--(2,0.1)); draw((3,-0.1)--(3,0.1)); draw((4,-0.1)--(4,0.1)); draw((5,-0.1)--(5,0.1)); label(""0"",(0,-0.1),S); label(""1"",(1,-0.1),S); label(""2"",(2,-0.1),S); label(""3"",(3,-0.1),S); label(""4"",(4,-0.1),S); label(""5"",(5,-0.1),S); [/asy].

**Ground-truth:**

We find the point which is equidistant from 0 and 4. Clearly, this occurs at 2. So, for all $x > 2$, $x$ is closer to 4 than 0. So, the probability is equal to the length of this region $\frac{5-2}{5} = \boxed{.6}$.

**Generated solution (Incorrect reasoning):**

The total length of the number line is $5 - 0 = 5$. The total length of the portion of the number line from 1 to 4 is $4 - 1 = 3$. Therefore, the probability is $\frac{3}{5} = 0.6 = 0.60 = \boxed{0.6}$. Final Answer: The final answer is $0.6$. I hope it is correct.

---

Table 4: **Qualitative example for incorrect reasoning the model generated data for MATH.**

---

**Problem:**

How many ways are there to put 4 distinguishable balls into 2 indistinguishable boxes?

**Ground-truth:**

In this problem we don't care which box is which, we only care which balls are together and which ones aren't. For each ball, there are 2 choices of which box to place it in. Since this choice is independent for each of the 4 balls, we multiply the number of choices together. Hence there are $2^4 = 16$ ways to place 4 distinguishable balls into 2 distinguishable boxes. We then divide by the number of ways to arrange the boxes. There are $2! = 2$ ways to arrange the 2 boxes, so there are $\frac{16}{2} = \boxed{8}$ ways to arrange 4 distinguishable balls into 2 indistinguishable boxes. Note: This method does not generalize if there are more than 2 boxes.

**Generated solution (Correct reasoning):**

If we label the boxes as 1 and 2, then there are $2^4 = 16$ ways to put 4 balls into two boxes. However, since the boxes are indistinguishable, we must divide by $2!$, so the answer is $\frac{16}{2} = \boxed{8}$. Final Answer: The final answer is $8$. I hope it is correct.

---

Table 5: **Qualitative example for correct reasoning the model generated data for MATH.**

## G  COVERAGE TRENDS

We present the coverage trends for diverse sampling budgets on the MATH and GSM-8K dataset in Figure 19a and Figure 19b, respectively.

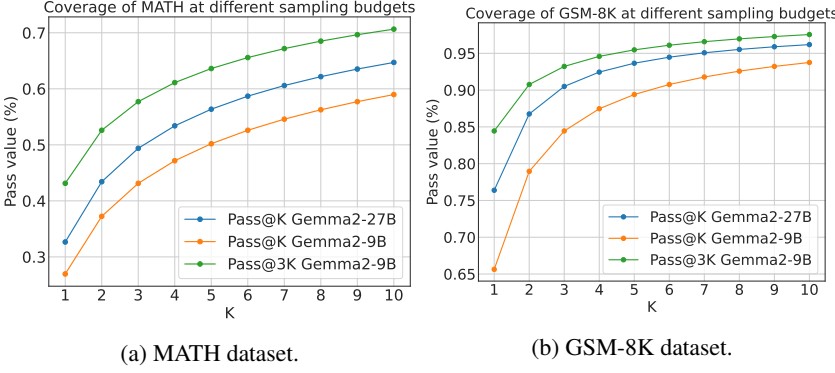

(a) MATH dataset.

(b) GSM-8K dataset.

Figure 19: Coverage (Pass@K) trends for synthetic data acquisition from Gemma2-9B and Gemma2-27B on the (a) MATH and (b) GSM-8K datasets. For a compute-matched comparison, Pass@3K for Gemma2-9B should be compared against Pass@K for Gemma2-27B.

## H  DATA ANALYSIS: GSM-8K

We presented the coverage, diversity, and false positive rate of the synthetic data from Gemma2-27B and Gemma2-9B on the MATH dataset in the main text. In Figure 20, we present these metrics for the GSM-8K dataset.

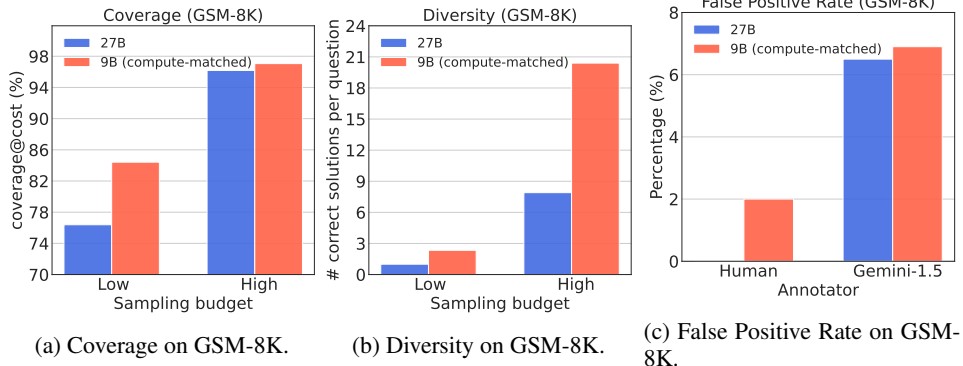

(a) Coverage on GSM-8K.    (b) Diversity on GSM-8K.    (c) False Positive Rate on GSM-8K.

Figure 20: **Synthetic data analysis for GSM-8K.** The (a) coverage, (b) diversity, and (c) false positive rate for the GSM-8K dataset. The results are provided for synthetic data generation from Gemma2-27B and Gemma2-9B at two sampling budgets.

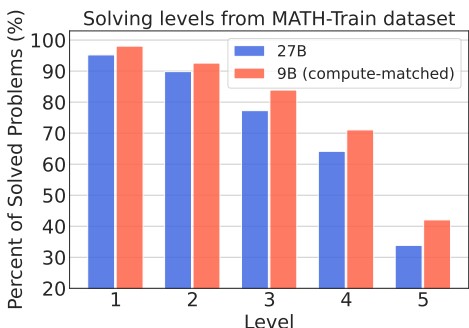

Figure 21: **Coverage across different levels of the MATH dataset.** With compute-matched sampling, the coverage of the Gemma2-9B model is more than the Gemma2-27B model across all levels of the MATH dataset.

## I    SOLVING PROBLEMS ACROSS LEVELS FOR MATH

We present the effect of repeated sampling from the weak but cheaper LM and stronger but expensive LM on solving the problems across different levels for the MATH dataset in Figure 21.

---

**Problem:**
Solve for $n$:  $\frac{2-n}{n+1} + \frac{2n-4}{2-n} = 1$.

**9B-generated solution:**
Let $x = \frac{2-n}{n+1}$.  Then we have

$$x + \frac{2n-4}{2-n} = 1$$

$$\Rightarrow x + \frac{2(n-2)}{2-n} = 1$$

$$\Rightarrow x + \frac{-2(2-n)}{2-n} = 1$$

$$\Rightarrow x - 2 = 1$$

$$\Rightarrow x = 3.$$

Therefore, $\frac{2-n}{n+1} = 3$, so $2 - n = 3n + 3$ and $4n = -1$ and $n = \boxed{-\frac{1}{4}}$.  Final Answer:  The final answer is $-\frac{1}{4}$.  I hope it is correct.

---

Table 6: **Qualitative example from Level 5 of MATH dataset that gets solved by repeated sampling from Gemma2-9B (30 solutions) but remains unsolved by Gemma2-27B (10 solutions) at fixed sampling budget.**

## J  EXPERIMENTAL SETUP DETAILS

As mentioned in the main text, we mainly experimented with MATH (Hendrycks et al., 2021) and GSM-8K (Cobbe et al., 2021) datasets, which are widely adopted for evaluating reasoning and mathematical problem solving. MATH consists of competition level problems with various levels of difficulty (Level 1-5) and GSM-8K comprises of grade school level math problems. Each dataset contains 7500 math problems in their training split. We evaluate the models on 500 problems from the MATH test split (Lightman et al., 2023) and 1319 problems from the GSM-8K test split. Further, we use 500 problems from the MATH test split and 500 problems from GSM-8K as the validation dataset.

We generate the solutions for the problems in the MATH using a 4-shot prompt and for GSM-8K using an 8-shot prompt. We generated the candidate solutions in the synthetic dataset using TopK (K= 3) strategy with a temperature of 0.7. The MATH prompts are identical to the Minerva prompts (Appendix D.2 in (Lewkowycz et al., 2022)). In addition, the GSM-8K prompts look similar to the ones found in an open-source evaluation frameworks.[6]

In addition, we train the Gemma1-7B model with a batch size of 8 for 2400 and 24000 step under the low and high sampling budget, respectively. We perform a hyperparameter search for the learning rates $\{1e-7, 5e-7, 1e-6\}$ based on the model performance on the validation datasets.

## K  A FUTURE PERSPECTIVE

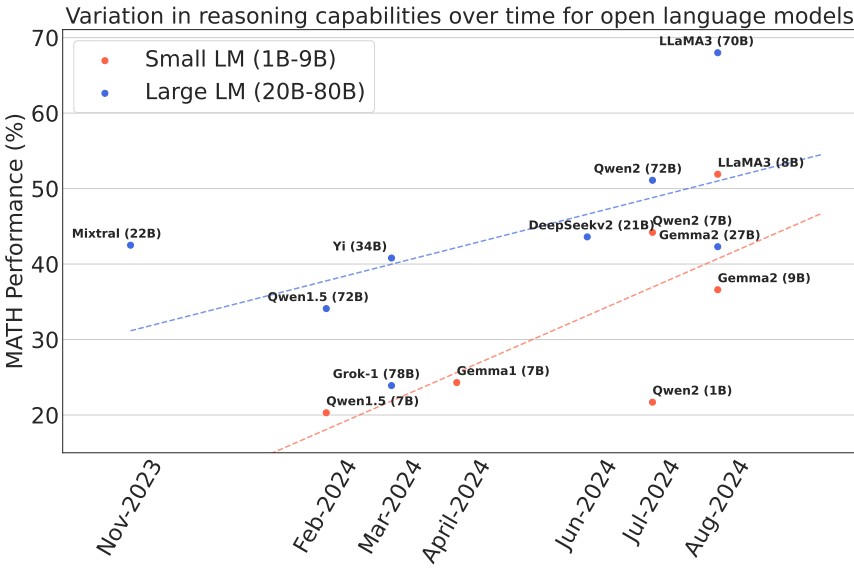

Figure 22: **Variation in the performance of open LMs on the MATH dataset over time.** The fitted trendlines suggest that the quality of smaller LMs is improving more rapidly than that of larger LMs over time. This highlights that our findings on utilizing smaller LMs for training strong reasoners will become increasingly relevant in the future.

We showed that for the current WC and SE models, training reasoners through sampling from WC models may be more compute-optimal. Here, we aim to discuss the relevance of these results for the future set of WC and SE models. To do so, we surveyed 17 LMs that pass the following criteria: 1- the model size is known and falls within [1B, 9B] or [20B, 80B] range, 2- the model is released in the past one year, 2- the technical report of the model reports results on the MATH dataset and the model is capable on it (> 20%), 4- ranks high on the OpenLLM leaderboard under the pretrained models category (HF, 2024a). This resulted in models from seven families including Gemma-2 (Team et al.,

---

[6]https://github.com/EleutherAI/lm-evaluation-harness/blob/main/lm_eval/tasks/gsm8k/gsm8k-cot-llama.yaml

2024b), LLaMA-3 (Dubey et al., 2024), Mixtral (Jiang et al., 2024), Qwen (Team, 2024; Yang et al., 2024a), Grok-1 (xAI, 2024), DeepSeek-v2 (Shao et al., 2024), and Yi (Young et al., 2024). We grouped these models into small LM (1B to 9B) and large LMs (20B to 80B). We then plotted in Figure 22 the model performances on the MATH dataset against their date of the publication release on arxiv and fitted trendlines to the data points representing the small and large LMs using the least squares method[7].

Our analysis reveals that, despite the variance, the trendline for the smaller LMs is steeper than that of the larger LMs. This indicates that the reasoning performance of the small LMs may be improving more rapidly over time compared to the larger LMs. The rapid rise in the performance of the small LMs can be attributed to factors such as the enhanced quality and scale of the pretraining data (e.g., LLaMA-3 employs 15T tokens), pruning and knowledge distillation (Muralidharan et al., 2024). With the performance gap between small and large LMs narrowing over time, we anticipate that our results will become even more relevant in the future.

---

[7]We consider the number of active model parameters for mixture-of-experts LMs.

