# OpenReview forum: "Smaller, Weaker, Yet Better: Training LLM Reasoners via Compute-Optimal Sampling"
_ICLR.cc/2025/Conference — ICLR 2025 Poster_

### Official Review · Reviewer_gC6v · 2024-10-22

**Soundness:** 4
**Presentation:** 4
**Contribution:** 2
**Rating:** 6
**Confidence:** 3

**Summary:**

The paper investigates whether it is better to (self-)distill from a Gemma-27B LLM or to distill three times more finetuning data from a three times smaller Gemma-9B model. It finds that the three times more data of the smaller model, despite including more errors, leads to a higher performance of the finetuned student model.

**Strengths:**

* The paper investigates not only knowledge distillation, but also a self-improvement setup
* In Figures 4 and 5, it is interesting that training on model-generated solution paths (one per question; at least in the "27B low compute" setup) gives a better performance than training on human-provided solution paths (also one per question)
* Figure 7 carries an interesting finding: Despite training on the data generated by the small model which has more errors, the ultimate trained model does not have more errors in its reasoning. This implies that the additional data mitigates its lower quality, which might add evidence to the discourse beyond the setup studied in this paper.
* The writing and flow of experiments is mostly clear

**Weaknesses:**

* In 21 of the 22 Figures, the paper hinges on "matching the compute", i.e., being allowed to generate 3 times more data when using the 3 times smaller LM. This confounds two factors of variation, making it hard to interpret the findings. This is the main weakness of the paper. One idea to improve on this weakness would be to test out distilling 1, 3, and 9 samples from the large LLM and 3, 9, and 27 samples from the small LLM (instead of the current 1+10 vs 3+30), so that there are both overlapping settings with a matched number of samples and with a matched compute.
* In the only figure where the small LLM is compared to the large LLM without this advantage (Figure 20 in the appendix), the large LLM produces better training data. It can be expected that if we use the large LLM to generate enough data until the student model converges, it will make a better distilled model. Thus, the only real application of the proposed method is when we do not have enough budget to produce enough data to converge. For the finetuning setup of the paper, that would amount to not being able to generate data for 8k-12.5k questions. This is a setup with limited applicability in practice. It would increase the contribution (score) of the paper to investigate problems where budget limits are hit more frequently in practice, like pretraining, see also my question below.
* Relative and absolute increases are reported inconsistently. E.g., in Figure 3b the fact that the proposed small model finds 11 instead of 5 solution paths per question (when it is allowed to generate 3 times more paths in total) is reported as a 125% increase (line 268), whereas the fact that 24% of its solutions paths are wrong compared to 17% of the large model is reported as a 7% increase (line 310). This inconsistency becomes problematic when reporting the increase on percentage numbers (e.g., line 258), where it is unclear whether this is a relative or absolute increase. Keeping the reporting consistent would increase both the presentation and the scientific soundness scores.
* The paper only evaluates Gemma (/Gemini) models. It would help judge the generalization of the claims (and increase the contribution score) to test it out on at least one other LLM, like a Llama model.
* The datasets are very limited to two math datasets, limiting the contribution. As above, more datasets would help judge the range of applicability, especially whether it also works on non-math and non-reasoning datasets.
* The paper does not compare to baselines, despite citing multiple closely related approaches
* The method still requires annotated data, because the LLM-generated data needs to be filtered out if it does not match the GT. It would increase the applicability of the score (and thus the contribution score) if there would be an ablation without filtering, i.e., answering whether the unfiltered erroneous data from the smaller model can still train a better model.

Small notes that did not influence my score and don't need to be rebuttled, I just note them to make the camera-ready better:
* The first paragraph of Section 3 could be shortened; it's message (in Equation 1) is just "if a model has x times more parameters, it takes x times longer to generate".
* typo in line 54, "filters"
* typo in line 103 "we supervise finetune"
* typo in line 151, "consists"
* typo in line 157, "for training student LM"
* typo in line 241, "that where"
* The references exclusively list arxiv versions of the papers, not their actual published versions
* The reference .bib file should best use double brackets for "{{OpenAI}}", "{{The Llama Team}}", to prevent the ill formatting in line 483 ("Team, 2024; Anthropic, 2024; AI, 2024")

**Questions:**

* Your distillation setup is limited to finetuning. One setup where it would be more realistic to not have enough budget is pretraining. Do you have any results on this? I of course do not expect to pretrain a network until convergence during the rebuttal, but it would already be helpful if you could show the first couple of iterations just to make sure the worse data (higher FPR) does not seem to converge to a much worse model.
* I'd be interested in sample-matched figures. The figures where I'd be most interested in a sample-matched comparison are Figures 4c and 5c. This would allow finding out if a small model can successfully improve a larger model, which would challenge beliefs in the field.
* Just to go sure: In the self-improvement setups, you keep training a model iteratively on its own generations from the current parameters? Or do you mean that you finetune a "fresh" 7B model using an already converged 7B model?

---

> ### Author Response · Authors · 2024-11-21
> **Response to reviewer**
>
> We thank the reviewer for their insightful comments. We are motivated to see that the reviewer finds our work (a) diverse in terms of the experimentation, (b) interesting its empirical findings, and (c) clearly written and easy to follow.
>
> Q: Matched number of samples vs matched compute
> - We believe there might be a misunderstanding with the overall objective of our work and we wish to clarify that.
> - Previous work has been comparing small and large models in what the reviewer calls “matched number of samples” setting, and found that data from the larger model is better. We also corroborate these results in Figure 18 of the paper by comparing small and large models in a number-matched setup (this is in addition to Figure 20 referenced by the reviewer and seems to be the result that the reviewer is asking for). Under a number-match setting, we also agree that it is better to use the large model.
> - However, we argue that the previous practice of comparing models in a number-matched setup has been putting the small model at a disadvantage because the small model spends substantially less compute compared to the large model during data generation. A fair comparison would give the same amount of compute to both the small and the large model.
> - We then compare small and large models in a fair setting where we allow both models the same amount of compute (i.e. compute-matched setup) and show that under a fair setting, the small model wins.
> - Important Note: From what we understand, the reviewer seems to consider the number-matched setting to be the fair setting, and the compute-matched setting to be a setting that gives the small model an advantage. However, our work argues that the compute-matched setting is the fair setting and the previous number-matched setting has been putting the small models at a disadvantage and not allowing for the full potential to be realized.
>
>
> Q: “that would amount to not being able to generate data for 8k-12.5k questions.”
> - We would like to also clarify that we are generating samples for all the 8k-12.5k questions. In fact, we are considering settings where we generate up to 30 samples per question with Gemma2 9B and 35 samples per question with Gemini 1.5 Flash. This amounts to O(0.5M) samples for finetuning.
>
> Q: Clarification regarding the confounding factors of variation
> - We clarify that the compute-matched sampling aims to sample N times more solutions from the small LM in comparison to the large LM where N is the ratio of the large LM and small LM capacity. For our experiments, we considered Gemma2-9B and 27B, hence, N = 3. Thus, we highlight that the multiplication factor is directly related to the model capacities rather than it being a confounder in our analysis.
> - Further, we clarify that some of our key results (Figure 1b and 8) also cover the state of the art language models i.e., Gemini-Flash and Pro. Here, we sampled 35x more solutions from the Flash model in comparison to the Pro model in the price-matched scenario. Hence, our experiments are applicable to open as well as closed language models.

---

> > ### Author Response · Authors · 2024-11-21
> > **Response to reviewer (2/n)**
> >
> > Q: Practical usefulness of the approach
> >
> > - We respectfully disagree with the reviewer’s take on this. Specifically, we clarify that GSM-8K and MATH datasets are the standard datasets in the language model reasoning literature [1,2,3].
> > - While the number of problems in these datasets may seem on the lower end (7.5K), as mentioned above we generate multiple solutions per problem for synthetic data creation. In particular, we generate 30 solutions and 10 solutions per problem from Gemma2-9B and Gemma2-27B for the two datasets, respectively, which amounts to 600K problem-solution pairs. In addition, we generate 35 solutions per problem from Gemini-Flash for the two datasets which amounts to 525K problem-solution pairs. We believe that this is quite large and at par with the size of the datasets used for supervised finetuning of the language models [4].
> > - Further, we clarify that the aim of the paper is not to reduce the cost of sampling data from the WC model, instead, we argue that it is more efficient use of your budget if you spend the “same” cost on sampling data from WC instead of the SE model for training language model reasoners. To strengthen our findings, we show that similar trends hold for instruction-following task too (Appendix B.2).
> > - We also highlight that the sampling budgets studied in our work are practically grounded with respect to some of the latest works in the literature. For instance, the order of solutions per problem from the MATH dataset in this recent work is around 30 [1].
> > - Finally, most practitioners and academics may have only a certain budget to sample data from large language models for supervised finetuning. In that regard, our paper proposes a practical solution to use this budget more optimally.
> >
> > [1] V-STaR: Training Verifiers for Self-Taught Reasoners: https://arxiv.org/abs/2402.06457 \
> > [2] STaR: https://arxiv.org/abs/2203.14465 \
> > [3] RestEM: https://arxiv.org/abs/2312.06585 \
> > [4] Alpaca: https://huggingface.co/datasets/tatsu-lab/alpaca
> >
> > Q: Robustness of conclusions
> >
> > - To address the reviewer’s comments in the limited time, we performed experiments with the Llama models. Specifically, for the MATH dataset, we generated 1 solution per problem from Llama-8B (SE) and 3 solutions per problem from Llama-3B (WC) in accordance with the compute-matched sampling ratio. Subsequently, we supervise finetuned Llama-1B, 3B, and 8B on the generated data from SE and WC models under the fixed sampling setup. We present the results on MATH500 test set below:
> >
> > | Data                       | Student-LM FT (LLama-1B)  | Weak-LM FT  (LLama-3B) | Strong-LM FT (LLama-8B) |
> > |----------------------------|---------------------------|------------------------|-------------------------|
> > | Llama-8B                   | 5.6                       | 31.6                   | 36.4                    |
> > | Llama-3B (compute-matched) | 7.2                       | 33.2                   | 38.2                    |
> >
> > - Consistent with our original results, we find that training with the WC data is more compute-optimal than SE data across diverse finetuning setups including knowledge distillation, self-improvement, and weak-to-strong improvement. We will add these results to the revised paper.
> > - Also consistent with our original results, we see that the WC model has a coverage of 67% and a diversity of 2.2, whereas the SE model has a coverage of 49% and a diversity of 1.
> > - We would also like to note that the models from the Gemma2 family, used in the original report, are quite capable open language models in their model capacity range. Specifically, Gemma2-9B achieves 36.6% in comparison to Mistral-7B’s 12.7% and LLaMA-2-7B’s 2.5% [1].  In addition, we experiment with state-of-the-art Gemini models (Pro/Flash) too and show that WC data outperform SE in this scenario. We believe these two families serve as solid evidence that our method works for open and closed language models, and the addition of the Llama results strengthen our claims even further.
> >
> > [1] Gemma2: https://arxiv.org/pdf/2408.00118

---

> > > ### Author Response · Authors · 2024-11-21
> > > **Response to reviewer (3/n)**
> > >
> > > Q: Addition of more tasks
> > >
> > > - We indeed go beyond math in Appendix B.2 and evaluate the usefulness of our approach in the context of instruction-following (chat) where the notion of final answer correctness is undefined. Specifically, we find that collecting price-matched responses from Gemini-Flash for a given instruction is more useful than acquiring responses from Gemini-Pro for training instruction-following language models. Hence, our work provides evidence that the WC sampling is possible in scenarios beyond reasoning tasks.
> > > - We also point out that the MATH and GSM datasets are standard datasets for studying language model reasoning with their wide adoption in the community.
> > > - Moreover, we evaluate our MATH-finetuned models on the Functional MATH dataset (Figure 6) to assess their generalization performance to distributionally-shifted math problems. We find that the compute-matched sampling from the small language models outperforms data from the large language models.
> > > - Although several complex evaluation reasoning tasks exist (e.g., GPQA), the absence of widely accepted training datasets restricts our ability to effectively fine-tune models for these tasks.
> > >
> > >
> > > Q:  Ablation without filtering
> > > - We have indeed extended our results to scenarios lacking ground-truth final answers (mentioned in L411, and Appendix B).
> > > - Specifically, we consider (a) no filtering, and (b) filtering using LM as a judge setting. Overall, the trends suggest that whether WC data is superior to SE data or not in the case of lacking ground truth data depends on the quality of the overall models and the finetuning setup.
> > > - Specifically, we find that compute-matched sampling from Gemini-Flash is more compute-optimal in comparison to Gemini-Pro in the absence of any filtering or using lm as a judge filtering for the MATH dataset.
> > > - Beyond reasoning, we consider instruction-following setup where there is no notion of filtering. In Figure 13, we find that WC sampling achieves a better IFEval score than SE sampling for instruction-following too.
> > >
> > >
> > > Q: Comparison to baselines
> > >
> > > - We are unclear which baselines the reviewer wants us to consider. The main baseline for work is to use the strong model to generate the samples, which we indeed compare to in all of our experiments.
> > > - In our related work section, we aim to discuss the contributions of several works that inspired or are contemporary to our work.
> > > - In addition, we consider supervised finetuning in this work which is the most widely adopted method to teach new skills to a language model since it is not resource heavy and technically straightforward.
> > >
> > > Q: Distillation and pretraining
> > >
> > > - Our work lays the foundation for revisiting the popular paradigm of using large language models for creating synthetic data. In its current state, our paper focuses on STaR [2], a popular strategy for building math reasoning models in the post-training stage using supervised finetuning.
> > > - While it will be interesting to study the implications of our method for pretraining, the experimental design is non-trivial. In particular, pretraining of language models requires a more complicated infrastructure due to the scale of tokens (trillions) and diversity of data domains (natural language, math, coding, multilingual data) involved in it. In comparison, the supervised finetuning is easier and more widely performed (at academic labs atleast) to teach new skills to the models. We will add this discussion in the revised paper.
> > >
> > > Q: Clarification on self-improvement setup
> > >
> > > - We clarify that the self-improvement experiments were performed on the Gemma2-9B and Gemma2-27B models. Akin to STaR [1], we first generate the solutions from the base model and finetune the same base model on the filtered data.
> > >
> > > [1] STaR: https://arxiv.org/abs/2203.14465
> > >
> > > Q: Consistent reporting of the scores and notes for minor edits
> > >
> > > - We thank the reviewer for pointing these out. We agree with the reviewer, and make the scores consistent throughout the paper and fix the minor edits in the revised version.

---

> ### Comment · Reviewer_gC6v · 2024-11-21
>
> I would like to thank the authors for the reply and especially the data on the Llama models. I just wanted to give an acknowledgement that I will respond in detail on Monday, since I am occupied with a parallel deadline until then.

---

> > ### Comment · Reviewer_gC6v · 2024-11-25
> >
> > Thank you for the reply and the additional experiments!
> >
> > > Q: Robustness of conclusions
> >
> > Thank you for the Llama results, which are exactly what I was asking for. I've increased my score as promised (see also below).
> >
> > > Q: Matched number of samples vs matched compute
> >
> > Thank you for pointing me to Figure 18, I indeed missed that one, since it was only mentioned in the context of mixture data in Appendix E.4. This is the figure I was asking for. Accordingly, (and in combination with the data on Llama), I'm increasing my soundness score to 4 and my total score to 6. Given that the paper is likely to be accepted, I would still encourage to discuss this result a bit more openly, since it is important to highlight for the connection to the broader field that your experiments support that the strong model data is still training better when training on the same number of distilled examples. The only places in the main text where I currently find this finding mentioned, are lines 58-60, line 177, and lines 367-369. For example in the last quote, _"Contrary to the common belief of self-generated data or data from a stronger model being better, our empirical findings show that training a model in a W2S-I setup from a WC data may be more compute-optimal than training it in a self-improvement setup on its own data"_, I agree that this is factually true. I just believe it would have provided more insight to me as a reader to formulate that "in line with previous findings (not contrary to it), strong models make better synthetic data -- but when changing the target to matching the compute it changes to the weak model being more efficient". I believe lines 367-369 would be good places to start this, possibly even an individual subsection (space permitting it), and especially in the discussion/conclusion. To me the current interpretation of the result nudges towards the idea that your paper is opposed to the previous literature, whereas I believe that it (beautifully) extends it, which is what the best science does and which is by no means a weakness if addressed openly. Addressing this final concern in the final camera-ready version (or promising to do so) would allow me to raise to a full 4/4 presentation score.
> >
> > > Q: Practical usefulness of the approach
> >
> > In the scenario of finetuning, following your numbers, finetuning one model on one dataset via the strong expensive models takes 75k samples. Assuming as an upper bound that we need 2000 tokens per answer, and that, as an upper bound, we use an expensive model like GPT4o, that's $1,500 (upper bound and just one-time cost). I agree it is not negligable, but it definitely is within budget to 3x this (and thereby outperform the weak cheap model). My point is that this paper's findings would become really impactful (and thus allow a higher contribution score) in settings where we cannot just increase the budget. Say pretraining, where we'd need to 1000x to get 75M samples. In this area, the paper's analysis would be more relevant to the field, because the weak-cheap model has no more real alternative (also it would address the issue that the amount of pretraining data scraped from the web is becoming a bottleneck currently). This is why I agree that this paper studies a niche problem very thoroughly (hence the soundness + presentation scores of 4 and the overall acceptance score of 6), but if the paper went into pretraining areas, it could easily be an 8+. I hope that this is a fair judgement that explains my overall score.
> >
> > > Specifically, we find that compute-matched sampling from Gemini-Flash is more compute-optimal in comparison to Gemini-Pro in the absence of any filtering
> >
> > Thank you for the experiment! It addresses my concern. It would help even more to have this for non-compute matched settings.
> >
> > > We clarify that the self-improvement experiments were performed on the Gemma2-9B and Gemma2-27B models. Akin to STaR [1], we first generate the solutions from the base model and finetune the same base model on the filtered data.
> >
> > Thank you for the clarification. It would be great to see this in the camera-ready paper (maybe even exactly as you wrote it here), since it is a more precise description than the current one in line 122 (_"Self-Improvement (Huang et al., 2022) corresponds to
> > training an LM on samples generated from itself."_) and 184-189.
> >
> > **TL;DR:** I'm increasing my soundness score (3 -> 4), presentation score (3 -> 4) and overall score (5 -> 6) as the new data and clarifications provided in the revised paper address many of my concerns. I am now leaning towards acceptance because the paper is now a thorough study. The reason I do not increase to 8 and that the contribution remains limited to 2 as the area of application is limited in my understanding. Hence, I see the paper as a scientifically thorough study of a niche problem. I hope that this judgement is fair and that my inputs on how to improve on the contribution score can help inspire follow-up works.

---

> > > ### Author Response · Authors · 2024-11-27
> > > **Response to the reviewer**
> > >
> > > We thank the reviewer for their thorough feedback and grateful for increasing their score.

---

### Official Review · Reviewer_qzpX · 2024-10-23

**Soundness:** 3
**Presentation:** 3
**Contribution:** 2
**Rating:** 6
**Confidence:** 4

**Summary:**

This paper presents the novel observation that generating synthetic data using a weaker but cheaper (WC) model is more effective than using a stronger but more expensive (SE) model. The authors demonstrate that, under the same budget, data generated by WC models tend to have higher coverage and diversity, though with a corresponding increase in false positive rates. Additionally, they show that models fine-tuned on WC-generated data consistently outperform those trained on SE-generated data.

**Strengths:**

The paper is well-structured and clearly written, making the methodology and results easy to follow.

The experiments are well-executed and provide convincing evidence of the benefits of the proposed approach.

It addresses a critical issue in synthetic data generation, offering a valuable contribution to this area of research.

**Weaknesses:**

The conclusion may not hold when using models from different companies. Based on my experience, under the same budget, data generated by a larger model like Qwen2.5 7B could outperform that of a smaller one like Gemma2 2B.

The paper could benefit from experimenting with more complex reasoning tasks, such as tree search algorithms, and using a reward model to evaluate the quality of the generated data.

**Questions:**

It seems that the difference in data quality between the WC and SE models becomes larger at lower budgets. Is it possible that the WC and SE models generate data of similar quality when the budget is very high?

---

> ### Author Response · Authors · 2024-11-20
> **Response to reviewer**
>
> We thank the reviewer for their diligent feedback. We are motivated to see that the reviewer finds our work: (a) well-structured and clearly written, (b) well-executed with convincing evidence, and (c) valuable contribution to the synthetic data generation research.
>
> Q: Robustness of the conclusions
>
> - We agree that models from different companies may have different properties making each of them suitable for a different use-case or task. While we do not provide results for comparing cross-company WC and SE models, our results so far help decide which model to use within the same family. For example, they help decide that within the Qwen2.5 family, one may be better of using a smaller model than the 7B model and within the Gemma2 family one may be better of using the 2B model instead of the larger models. Extending our results to models from different companies and understanding when it works as well as the failure modes is a great future direction.
> - Note that the aims of the current work is to lay the foundation for compute-matched sampling to train LM reasoners. We perform experiments with WC and SE models from the same model family which is not an unreasonable assumption. In our paper, we show that WC data outperforms SE data from Gemma (open) as well as Gemini (closed) language models. We further point out that the Gemma and Gemini series models are quite different from each other based on the information available publicly. For instance, Gemma [1] models are purely text language models while Gemini models are natively multimodal in nature which would lead to an entirely different learned data distribution [2].
> - To provide further evidence, we performed experiments with the Llama models. Specifically, for the MATH dataset, we generated 1 solution per problem from Llama-8B (SE) and 3 solutions per problem from Llama-3B (WC) in accordance with the compute-matched sampling ratio. Subsequently, we supervise finetuned Llama-1B, 3B, and 8B on the generated data from SE and WC models under the fixed sampling setup. We present the results on MATH500 test set below:
>
> |            Data            | Student-LM FT (LLama-1B)  | Weak-LM FT  (LLama-3B) | Strong-LM FT (LLama-8B) |
> |:--------------------------:|:-------------------------:|:----------------------:|:-----------------------:|
> |          Llama-8B          |            5.6            |          31.6          |           36.4          |
> | Llama-3B (compute-matched) |            7.2            |          33.2          |           38.2          |
>
> - Consistent with our original results, we find that training with the WC data is more compute-optimal than SE data across diverse finetuning setups including knowledge distillation, self-improvement, and weak-to-strong improvement. We will add these results to the revised paper.
> - Also consistent with our original results, we see that the WC model has a coverage of 67% and a diversity of 2.2, whereas the SE model has a coverage of 49% and a diversity of 1.
>
> - We leave the expansion to diverse pairs of the (WC, SE) models for the future work. We will add this discussion in Appendix A.
>
> [1] Gemma2: https://arxiv.org/pdf/2408.00118 \
> [2] Gemini 1.5: https://storage.googleapis.com/deepmind-media/gemini/gemini_v1_5_report.pdf

---

> > ### Author Response · Authors · 2024-11-20
> > **Response to reviewer (2/n)**
> >
> > Q: Experiments on more tasks/datasets
> > - We have indeed extended our results to scenarios lacking ground-truth final answers (mentioned in L411, and Appendix B) by using reward models (i.e. verifiers) instead. Specifically, we consider (a) no filtering, and (b) reward modeling using LM as a judge setting. In the latter case, we propose a framework for keeping the computations the same despite using a reward model for each of the samples generated by the models. Overall, the trends suggest that whether WC data is superior to SE data or not in the case of lacking ground truth data depends on the quality of the overall models and the finetuning setup. Our framework can be used with other more sophisticated reward models by simply replacing the LM as a judge with those reward models.
> > - We point out that the MATH and GSM datasets are standard datasets for studying language model reasoning with their wide adoption in the community. Although several complex evaluation reasoning tasks exist (e.g., GPQA), the absence of widely accepted training datasets restricts our ability to effectively fine-tune models for these tasks.
> > - Specifically, we find that compute-matched sampling from Gemini-Flash is more compute-optimal in comparison to Gemini-Pro in the absence of any filtering or using lm as a judge reward modeling for the MATH dataset.
> > - Beyond reasoning, we consider instruction-following setup where there is no notion of filtering. In Figure 13, we find that WC sampling achieves a better IFEval score than SE sampling for instruction-following too.
> > - Extending our work to sampling with tree search is a great direction for future research, but we believe that is a direction that warrants a separate publication with extensive results, and out of scope for this work.
> >
> > Q: Extremely high budget
> >
> > - Firstly, we highlight that the sampling budgets studied in our work are practically grounded with respect to some of the latest works in the literature. For instance, the order of solutions per problem from the MATH dataset in this recent work is around 30 [1].
> > - Secondly, most practitioners and academics may have only a certain budget to sample data from large language models for supervised finetuning. In that regard, our paper proposes a practical solution to use this budget more optimally.
> > - In our early experiments with a very high budget, we found that there was no difference between the performance of models finetuned with SE and WC models. We believe that the high false positive rate of the WC model becomes more prominent and hurts the performance. However, we leave a deeper study of this phenomenon as a future work.
> >
> > [1] REST-EM: https://arxiv.org/pdf/2312.06585

---

> > > ### Author Response · Authors · 2024-11-23
> > > **Rebuttal reminder**
> > >
> > > Hi,
> > >
> > > Thanks again for your insightful feedback on our work! We've carefully worked to address your comments/questions. Are there any further questions or concerns we should discuss?

---

### Official Review · Reviewer_rLZh · 2024-10-28

**Soundness:** 3
**Presentation:** 3
**Contribution:** 3
**Rating:** 8
**Confidence:** 3

**Summary:**

This paper revisits the trade-offs between generating synthetic data using a stronger but more expensive (SE) model versus a weaker but cheaper (WC) model, and finds that at a fixed sampling compute budget, finetuning LMs with data from a WC model can consistently outperform data from a SE model in multiple settings.

**Strengths:**

1. The research question is significant, focusing on performance comparison of data sampled from WC and SE models, respectively.
2. The findings are impressive, challenging the traditional belief that data from a strong model is better for finetuning models.
3. The evaluation settings are diverse, demonstrating the effectiveness and robustness of this method, despite only the Gemma series models.

**Weaknesses:**

1. This paper centers exclusively on the Gemma series, and it is essential to extend the analysis to the Llama series to demonstrate the robustness of the conclusions.
2. While the paper aims to highlight the lower computational cost of the WC model for data synthesis (particularly important for large-scale data generation), all the experiments are conducted on relatively small datasets. This discrepancy undermines the overall contribution of the paper.
3. Compared to the SE model, the WC model can be regarded as a more diverse yet lower-quality variant. Therefore, it is crucial to compare it with techniques designed to enhance output diversity. Specifically, if adjusting the sampling temperature of the SE model consistently results in performance degradation relative to the WC model, this suggests that the WC model provides a superior quality-diversity trade-off compared to merely increasing the sampling temperature.

**Questions:**

1. Although both low and high budgets are studied, could you please provide the results of an extremely high budget where the cost is not an important factor? This should be indicative of diverse data scales.
2. Despite the train-test splits of MBPP, this paper only trains models on MBPP and tests them on HumanEval. The testing results on MBPP are expected to be provided for a more comprehensive understanding.
3. Writing：
- 1. All the ref links are invalid.
  2. l101, l104: "i.e." should be "i.e.".
  3. l103: grammar error for "we supervise finetune".
  4. l109: use \citet{} for "(Zelikman et al., 2024;Singh et al., 2023).".

---

> ### Author Response · Authors · 2024-11-20
> **Response to reviewer**
>
> We thank the reviewer for their insightful comments. We are happy to note that the reviewer finds our work (a) significant in comparing SE and WC models, (b) impressive in terms of the findings that challenge the traditional beliefs, and (c) effective and robust in terms of the evaluation.
>
> Q: Robustness of conclusions
>
> - Thanks for the suggestion. To address the reviewer’s comments in the limited time, we performed experiments with the Llama models. Specifically, for the MATH dataset, we generated 1 solution per problem from Llama-8B (SE) and 3 solutions per problem from Llama-3B (WC) in accordance with the compute-matched sampling ratio. Subsequently, we supervise finetuned Llama-1B, 3B, and 8B on the generated data from SE and WC models under the fixed sampling setup. We present the results on MATH500 test set below:
>
> |            Data            | Student-LM FT (LLama-1B)  | Weak-LM FT  (LLama-3B) | Strong-LM FT (LLama-8B) |
> |:--------------------------:|:-------------------------:|:----------------------:|:-----------------------:|
> |          Llama-8B          |            5.6            |          31.6          |           36.4          |
> | Llama-3B (compute-matched) |            7.2            |          33.2          |           38.2          |
>
> - Consistent with our original results, we find that training with the WC data is more compute-optimal than SE data across diverse finetuning setups including knowledge distillation, self-improvement, and weak-to-strong improvement. We will add these results to the revised paper.
> - Also consistent with our original results, we see that the WC model has a coverage of 67% and a diversity of 2.2, whereas the SE model has a coverage of 49% and a diversity of 1.
> - We would like to also note that the models from the Gemma2 family, used in the original report, are quite capable open language models in their model capacity range. Specifically, Gemma2-9B achieves 36.6% in comparison to Mistral-7B’s 12.7% and LLaMA-2-7B’s 2.5% [1].  In addition, we experiment with state-of-the-art Gemini models (Pro/Flash) too and show that WC data outperform SE in this scenario. We believe these two families serve as solid evidence that our method works for open and closed language models, and the addition of the Llama results strengthen our claims even further.
>
> [1] Gemma2: https://arxiv.org/pdf/2408.00118
>
> Q: Scale of data collection
>
> - We respectfully disagree with the reviewer’s take on this. We would like to clarify two important points: \
> 1- While the number of problems in our datasets may seem on the lower end (7.5K), we generate multiple solutions per problem for synthetic data creation. In particular, we generate 30 solutions and 10 solutions per problem from Gemma2-9B and Gemma2-27B for the two datasets, respectively, which amounts to 600K problem-solution pairs. In addition, we generate 35 solutions per problem from Gemini-Flash for the two datasets which amounts to 525K problem-solution pairs. We believe that this is quite large and at par with the size of the datasets used for supervised finetuning of the language models [4]. \
> 2- Perhaps more importantly, we would like to clarify that the aim of the paper is NOT to reduce the cost of sampling data and/or make the approach scale to larger datasets. Instead, we argue that given a fixed budget (i.e. the budget one is willing to spend for training a model, in terms of compute or cost), it is more efficient usage of the budget if one spends it on sampling data from WC instead of the SE model for training language model reasoners as well as instruction following models. As a particular example, imagine one wants to spend 1000USD on tuning a model on their training data, and they have the choice of spending it on sampling from a WC or a SE model. Our results suggest that no matter how small or large their training data is, if they spend the 1000USD sampling from WC, they may end up with a better model than if they spend it sampling from SE.
> - We also note that GSM-8K and MATH datasets are standard datasets in the language model reasoning literature [1,2,3].
>
> [1] V-STaR: Training Verifiers for Self-Taught Reasoners: https://arxiv.org/abs/2402.06457 \
> [2] STaR: https://arxiv.org/abs/2203.14465 \
> [3] RestEM: https://arxiv.org/abs/2312.06585 \
> [4] Alpaca: https://huggingface.co/datasets/tatsu-lab/alpaca
>
> Q: Quality-diversity with WC and SE model
> - In our early experiments, we found that changing the sampling temperatures from 0.7 (the one used for the final experiments) to either 0.5 (i.e. lower) or 1.0 (i.e. higher) would lead to lower performance than 0.7 for both WC and SE data. This finding corroborates the reviewer’s point that the WC model may be providing a superior quality-diversity trade-off compared to merely increasing the sampling temperature.

---

> > ### Author Response · Authors · 2024-11-20
> > **Response to reviewer (2/n)**
> >
> > Q: Extremely high budget
> > - Firstly, we highlight that the sampling budgets studied in our work are practically grounded with respect to some of the latest works in the literature. For instance, the order of solutions per problem from the MATH dataset in this recent work is around 30 [1].
> > - Secondly, most practitioners and academics may have only a certain budget to sample data from large language models for supervised finetuning. In that regard, our paper proposes a practical solution to use this budget more optimally.
> > - In our early experiments with a very high budget, we found that there was no difference between the performance of models finetuned with SE and WC models. We believe that the high false positive rate of the WC model becomes more prominent and hurts the performance. However, we leave a deeper study of this phenomenon as a future work.
> >
> > [1] REST-EM: https://arxiv.org/pdf/2312.06585
> >
> > Q: MBPP
> >
> > - We highlight that MBPP-sanitized is an extremely small dataset consisting of 427 problems. We used 3 of them for few-shot prompting, 324 for training the models, and 100 problems for validation. There is no dedicated test split that comes with the dataset. Even if we were to create a small test set out of 427 problems, it would come at the cost of reducing the number of training examples and noise in our testing performance estimations.
> > - To solve this problem, we perform our analysis on the out of distribution dataset (HumanEval) which is a widely adopted evaluation dataset in the community.
> >
> > Q: Writing
> >
> > - Thank you for pointing out the writing errors. We will fix them in the revised version of the paper.

---

> > > ### Author Response · Authors · 2024-11-23
> > > **Rebuttal reminder**
> > >
> > > Hi,
> > >
> > > Thanks again for your insightful feedback on our work! We've carefully worked to address your comments/questions. Are there any further questions or concerns we should discuss?

---

> > > > ### Comment · Reviewer_rLZh · 2024-11-26
> > > > **Response to Authors**
> > > >
> > > > Sorry for the late reply! I still believe that this topic is interesting, and will inspire the expenditure of data generation. Based on your responses, there are only three following questions, or more like open discussions. If resolved, I would like to consider raising my rating.
> > > >
> > > > 1. Since the discrepancy between WC and SE models will disappear with the scaling of generated data, I recommend discussing this point in your paper. Two curves in one figure to illustrate this trend would be invaluable for readers. Currently, the prevailing message seems to be the preference for a WC model in any scenario.
> > > >
> > > > 2. Trade-off between the validity and diversity of data: Do you observe a minimum viable size for the WC model? Specifically, a much smaller WC model than the current one would generate low-quality data, even if it can yield more with the same budget. As indicated by the results at higher sampling temperatures, lower data quality will lead to higher FPR, and impede performance improvements. This issue may worsen as task difficulty increases.
> > > >
> > > > 3. Would using a WC model cause more finetuning overhead due to more generated data? If so, how much is the overhead?

---

> > > > > ### Author Response · Authors · 2024-11-27
> > > > > **Response to the reviewer**
> > > > >
> > > > > We thank the reviewer for their questions.
> > > > >
> > > > > 1. We clarify that the usefulness of the WC over SE data emerges from better coverage and diversity. With infinite sampling budget, one or both of these factors would not be different for the WC and SE data and false positive rates might become more prominent. Such trends might change with the tasks at hand too. As suggested by the reviewer, we will add this message to the updated paper for the readers to understand the possible limitations of the work.
> > > > >
> > > > > 2. Good point! We agree that a very small WC model will not be able to solve difficult tasks or generate very low quality solutions. In our early experiments, we had found that Gemma2-2B's coverage increases significantly with more sampling but the FPR was quite high. That is why, we study the axis of coverage, diversity, and FPR to understand the benefit of WC data over SE data.
> > > > >
> > > > > 3. In our experiments, the models are finetuned with the same number of finetuning steps, and we save multiple checkpoints during the run. Subsequently, we pick the best checkpoint based on the performance on the validation data (L208-211 in the revised draft).
> > > > >
> > > > > We thank the reviewer again for their questions! We hope our response gives more confidence in our work. Feel free to ask more questions.

---

> > > > > > ### Comment · Reviewer_rLZh · 2024-11-28
> > > > > > **Response to Authors**
> > > > > >
> > > > > > I've rapidly looked through the questions and responses from all the reviewers. The authors have throughly resolved my concerns, and convinced me to raise the score from 6 to 8, based on the understanding that authors will incorporate the valuable feedback, especially the potential limitations, into their camera-ready version.
> > > > > >
> > > > > > Due to the chaotic reviews this year, I hope that we could still follow the recommendations to uphold the reputation of the ICLR community!

---

### Official Review · Reviewer_6hvc · 2024-11-02

**Soundness:** 3
**Presentation:** 2
**Contribution:** 3
**Rating:** 8
**Confidence:** 3

**Summary:**

This paper challenges the common practice of using strong but expensive (SE) language models to generate synthetic training data, proposing instead that using weaker but cheaper (WC) models may be more compute-optimal. The authors introduce a "compute-matched sampling" framework that enables fair comparison between WC and SE models by accounting for their relative compute costs. At a fixed compute budget, this framework shows that one can generate P_SE/P_WC more samples from a WC model than an SE model. The authors evaluate this approach across multiple model pairs (Gemma2 9B/27B and Gemini Flash/Pro), tasks (primarily mathematical reasoning), and training paradigms (knowledge distillation, self-improvement, and a novel "weak-to-strong improvement"). They assess the generated data along three key dimensions: coverage (problems solved), diversity (unique solutions per problem), and false positive rate (correct answers with incorrect reasoning). The results consistently show that training with WC-generated data outperforms SE-generated data when properly compute-matched.

**Strengths:**

1. _Originality_:
   - Introduces a novel compute-matched sampling framework with clear mathematical foundations
   - Proposes a new "weak-to-strong improvement" training paradigm that challenges conventional wisdom
   - Provides a fresh perspective on the compute-quality trade-off in synthetic data generation

2. _Experimental Rigour_:
   - Comprehensive evaluation across multiple dimensions:
     * Multiple model pairs (both open and closed models)
     * Various compute budgets and training paradigms
     * Different dataset sizes and difficulty levels
   - Thorough ablation studies that isolate the impact of coverage and diversity
   - Both human and automated evaluation of false positive rates
   - Clear validation of results through transfer learning (Functional MATH)

3. _Practical Impact_:
   - Demonstrates significant cost savings potential (0.15x cost for comparable or better performance)
   - Shows consistent improvements across model sizes (7B to 27B)
   - Provides actionable insights for practitioners
   - Results particularly relevant given the trend of improving smaller models

4. _Technical Depth_:
   - Rigorous mathematical formulation of compute-matching
   - Analysis of traed-offs between coverage, diversity, and error rates
   - Ablation studies support main claims
   - Clear empirical validation of theoretical framework

**Weaknesses:**

1. _Theoretical Foundation_:
   - Lacks formal analysis of when WC sampling should outperform SE sampling
   - No theoretical bounds on the optimal sampling ratio
   - Missing analysis of the relationship between model size and optimal sampling strategy
   - Limited exploration of failure modes and their characteristics

2. _Methodology Limitations_:
   - Heavy reliance on ground truth for filtering solutions
   - Limited exploration of alternative filtering strategies
   - FPR evaluation methodology could be more robust (50 human samples probably insufficient)
   - Some key implementation details relegated to appendices

3. _Generalisation Concerns_:
   - Primary focus on mathematical reasoning tasks
   - Limited exploration of other domains (coding results show context-dependency)
   - Unclear scalability to larger model sizes
   - Performance on more complex reasoning tasks not fully explored

4. _Practical Considerations_:
   - Deployment challenges in scenarios without ground truth not fully addressed
   - Resource optimisation strategies could be explored more
   - Limited discussion of integration with existing training pipelines
   - Cost-benefit analysis could be more comprehensive across different scenarios

**Questions:**

I will try and cluster my questions in sensible groups.

1. _Theoretical Understanding_:
   - Can you provide theoretical insights into when WC sampling should outperform SE sampling?
   - How does the optimal sampling ratio change with model size and task complexity?
   - What are the key factors that determine the success of weak-to-strong improvement?

2. _Methodology_:
   - How would the results change with more sophisticated filtering strategies?
   - Could you provide more details about the specific prompting strategies used?
   - How sensitive are the results to the choice of temperature and sampling parameters?

3. _Generalisation_:
   - What characteristics of a task make it more/less suitable for WC sampling?
   - How would the results scale to even larger model sizes?
   - What is the relationship between FPR and final model performance?

4. _Practical Implementation_:
   - How would you recommend implementing this in scenarios without ground truth?
   - What modifications would be needed for different domains or tasks?
   - Could you provide more detailed guidance on optimal sampling strategies for different scenarios?

---

> ### Author Response · Authors · 2024-11-17
> **Response to reviewer**
>
> We thank the reviewer for their diligent feedback. We are motivated to see that the reviewer finds our work (a) novel with a fresh perspective on synthetic data generation, (b) experimentally rigorous, (c) practically impactful, and (d) rigorous in analysis and experimental validation.
>
> Q: Reliance on ground truth, filtering strategies, and more tasks.
>
> - We agree these are important directions for extending our work and we have indeed included experimental results both for 1- different filtering strategies beyond ground truth labels, and for 2- tasks beyond math and coding. These results are referenced on lines 407-413 of the main text, with the results being presented in the appendix due to space limitations. If the reviewer finds these parts of our result more interesting than any other part in the main text, we are happy to move them to the main text. In what follows, we provide a high-level summary of the results.
> - We extend our results to scenarios lacking ground-truth final answers (mentioned in L411, and Appendix B). Specifically, we consider (a) no filtering, and (b) filtering using LM as a judge setting. In the latter case, we propose an approach to still match the computes, despite using an LM for judging model generated answers. Overall, the trends suggest that whether WC data is superior to SE data depends on the quality of the overall models and the finetuning setup in both the settings. In particular, we notice that Gemini-Flash generated data still outperforms Gemini-Pro generated data without access to ground-truth data.  Future work can extend our results by using trained verifiers instead of llm-as-a-judge, in a similar compute-matched setup.
> - Beyond reasoning, we consider the instruction-following setup where there is no notion of filtering. In Figure 13, we find that WC sampling achieves a better IFeval score than SE sampling for instruction-following too, thus showing that our results can be extended beyond reasoning.
> - Lastly, we clarify that it is quite common to assume access to the ground-truth final answer in the LLM reasoning literature [1,2,3]. Specifically, many real-world datasets such as MATH and GSM-8K come with final answers, and many coding datasets come with test-cases that can be used to judge the correctness of a generated code. Note that in our work, we do not use human-written chain-of-thoughts to solve the math problems, we just utilize the final answers to filter the chain of thoughts that lead to incorrect final answers.

---

> ### Author Response · Authors · 2024-11-17
> **Response to reviewer (2/n)**
>
> Q: Insights into WC and SE sampling.
>
> - In this work, we lay the foundation for this direction by showing that, perhaps surprisingly and contrary to the common belief, data from smaller LMs can be more compute-optimal than data from larger MLs. As the reviewer mentioned, our work opens up the avenue for a great body of theoretical analysis on when WC may be superior than SE.
> - So far, our understanding based on the experiments reported in the current work is that whether a WC can outperform a SE depends on how they compare in terms of coverage, diversity, and false positive rate (FPR). In particular, we believe the FPR of the WC model should be low (as evidenced by our experiments with filterings other than ground truth), and the coverage of the WC model should be higher than SE (when the two models have similar coverage, they tend to lead to similar performance).
> - To get a more nuanced understanding of when WC outperforms SE, one possible experimental design is to compute the coverage, diversity, FPR, and potentially other features for several models and several datasets, and then finetune models on data from these models and compute the delta in their performance. Then fit lines/curves that predict the delta in performance with respect to the three properties. The weights assigned to these features can then be indicative of how important each feature is with respect to other features and allow for predicting whether a WC may outperform an SE only based on these features, before running any finetuning experiments. This is, however, quite computationally demanding and beyond the scope of the current work. We hope future work will dive deeper into this.
> - We point that we study the notion of (a) coverage, (b) diversity, and (c) false positive rates to get insights into WC and SE sampling from Gemma and Gemini models (L209-237).
> - Our experiments reveal that the WC sampling achieves higher coverage and diversity than SE sampling while SE sampling achieves a lower false positive rate (FPR) at the fixed sampling budget. Subsequently, our supervised finetuning results indicate that WC sampling can achieve better scores than SE sampling.
> - While it is pertinent, we mention in our discussion (Appendix A) that the aim of our work is not to come up with the conditions under which WC sampling will outperform SE sampling.
> - To achieve this, we will need to curate a large number of datasets with diverse coverage, diversity and FPR values and run many finetuning runs subsequently. However, this paper lays foundations for this future exploration.
>
> Q: Performance on more complex reasoning tasks not fully explored.
>
> - In this work, we consider the MATH reasoning dataset as our main testbed, which is standard in most of the LLM reasoning papers [3,6]. This dataset contains more complex problems in comparison to the high-school level GSM-8K dataset.
> - In Figure 22, we show that the compute-matched sampling from the WC model increases the coverage across all the levels i.e., (easy: level 1 to complex: level 5).
> - In addition, we evaluate our MATH-finetuned models on the Functional MATH dataset (Figure 6) to assess their generalization performance to distributionally-shifted problems. Note that Functional MATH has been shown to be quite challenging, even for the frontier models.
> - Although several complex evaluation reasoning tasks exist (e.g., GPQA), the absence of widely accepted training datasets restricts our ability to effectively fine-tune models for these tasks.
>
> Q: Theoretical analysis
>
> - While we do have non-theoretical insights for now: the higher the coverage and diversity of WC and the lower the FPR, the more chance WC will work.
> - In practice, this theoretical analysis will include many models with different coverage, diversity, and FPR. Subsequently, we can try to fit curves that predict the model performance.
> - Then, the weights of these variables could tell us how important they are with respect to each other. However, this will require a lot of compute, which is out of scope of this work.

---

> > ### Author Response · Authors · 2024-11-17
> > **Response to reviewer (3/n)**
> >
> > Q: Optimal sampling ratio with model size and task complexity.
> >
> > - In this work, we consider synthetic data generation at a fixed sampling budget (L138-140), where the sampling ratio between weak and cheap (WC) and strong and expensive (SE) model is the ratio of their model sizes (capacity). The ratio is only used for comparing two models. In practice, there will be no notion of “optimal ratio”. Depending on how much compute one is willing to spend on sampling, one can determine how many samples from each of their models they can generate.
> > - In Figure 1, we consider two sampling ratios: 3x for Gemma2-9B and Gemma2-27B model and 35x for Gemini-Flash and Gemini-Pro. Our experiments reveal that compute-matched sampling from the WC model is a better allocation of sampling resources than the SE model.
> > - Further, our experiments show that the compute-matching sampling is more optimal than number-matched sampling (Figure 18 from Appendix D.3).
> > - In this work, we do not vary the sampling ratio with the task-complexity but it is an interesting dimension that warrants further exploration. We will add this point in our discussion (Appendix A) section.
> >
> > Q: Factors behind the success of weak-to-strong improvement.
> >
> > - In this work, we cover three dimensions to assess the synthetic data quality – coverage, diversity, and false positive rates. Similar to the other setups, we believe high coverage and diversity and low false positive rate would be desired for the success of weak-to-strong improvement.
> > - As the quality of the WC models improve (Figure 9), we believe that coverage, diversity and FPR will improve.
> > - Additionally, the development of high-quality verifiers will be essential to filter bad chain-of-thoughts from the synthetic data. We will add this discussion in the revised paper.
> >
> > Q: Prompting strategies.
> > - As mentioned in L1232, we use a 4-shot prompt to generate solutions for the MATH dataset. Specifically, these prompts are identical to the Minerva prompts (Appendix D.2 in [4]).
> > - In addition, we use a 8-shot prompt to generate solutions for the GSM-8K dataset. Those prompts look similar to the ones found in popular open-source evaluation frameworks [5]. We will update the appendix to include those prompts for completeness.
> >
> > Q: Temperature and sampling parameters.
> >
> > - Consistent with prior works [3], we chose the sampling temperature of 0.7 (L 1234). As a standard in many LLM inference pipelines, we use the topK sampling as our default strategy.
> > - In our early experiments, we had observed that using a sampling temperature of 0.7 is better than using temperature of 0.5 and 1.0 to achieve higher coverage on MATH problems as it balances consistency and creativity in the solutions. This observation is similar to other works on STaR-style training, and not specific to our work.
> >
> > Q: FPR evaluation methodology.
> >
> > - We agree that the human evaluation of the false positive rates is limited due to time and budget constraints for human assessments. We will mention this explicitly in the discussion (Appendix A) section.
> > - However, we observed that the trends from the human evaluation were consistent with large-scale automatic evaluation with a larger language model (e.g., Gemini-1.5 Pro), as shown in Figure 3c.
> >
> > Q:  Characteristics of a task that make it more/less suitable for WC sampling.
> >
> > - Our work shows that WC sampling can be suitable both for reasoning and instruction following tasks. In general, as evidenced by our experiments with no filtering or LM as a judge filtering, we expect that the tasks that benefit more from WC are those for which either the WC has a good quality without any filtering, or we have access to powerful verifiers that enable us to remove lower quality model outputs.
> >
> > Q: Scaling to even larger model sizes.
> >
> > - In this paper, we consider synthetic data generation from a wide range of models: open models such as Gemma-9B, 27B and state-of-the-art LMs such as Gemini-Flash and Pro.
> > - In addition, we consider finetuning models of diverse model sizes: 7B, 9B and 27B.
> > - We firmly believe that the model sizes in our experiments are a good representative of the academic (open) and the industrial standards (API-based language models).

---

> > > ### Author Response · Authors · 2024-11-17
> > > **Response to reviewer (4/n)**
> > >
> > > Q: Relationship between FPR and final model performance?
> > >
> > > - We clarify that the FPR measures the quality of the generated chain-of-thought i.e., whether we achieve the correct answer with accurate reasoning. On the other hand, the final model performance measures whether the generated solution leads to the correct final answer. Ideally, we want to maximize the model performance and minimize the FPR.
> > > - In our experiments, we observed that the WC sampling achieves a higher model performance (Figure 4) than SE sampling while the FPR of the finetuned models is roughly the same for both the models (Figure 7, L366-374).
> > > - Since FPR is hard to determine with high precision, we do not provide an exact relationship between FPR and final model performance across many runs. We leave a deeper investigation into this as a future work.
> > >
> > > Q: Modifications needed for different domains or tasks.
> > >
> > > - We do not expect major differences or modifications being needed for extending our results to different domains or tasks. While we started our experiments for math datasets, we found it quite straightforward to extend our results to code and instruction following tasks, with the only modification needed being to change the filtering approach (we used no filtering for instruction following, and test case passing for code).
> > > - For a new domain and task, we recommend assessing whether the coverage and diversity is high while the FPR is low with WC sampling. Overall, our paper finds that these characteristics lead to good model performance.
> > >
> > >
> > > Q: Discussion of integration with existing training pipelines
> > > - Our work lays the foundation for revisiting the popular paradigm of using large language models for creating synthetic data. In its current state, our paper focuses on STaR [2], a popular strategy for building math reasoning models in the post-training stage.
> > > - Since the setups considered in our paper are relatively straightforward, we believe that it should be easy for anyone to replace their large language models with their smaller versions to perform compute-matched sampling.
> > > - In addition, we perform supervised finetuning for training the language models which is the most widely popular strategy to teach new skills to the model (e.g., math reasoning).
> > >
> > >
> > > Q: Cost-benefit analysis across different scenarios
> > >
> > > - Our main results suggest that it is more cost-efficient to sample synthetic data from WC models (Gemma2-9B) in comparison to SE models (Gemma2-27B) for math reasoning tasks, at the fixed sampling budget in terms of FLOPs.
> > > - We find similar results when you do not have access to the FLOPs for the closed language models. In that case, we perform price-matched sampling from WC (Gemini-Flash) and show that it is more cost-effective than sampling from SE (Gemini-Pro) for math reasoning tasks.
> > > - We expand our main results to scenarios where we spend less money on sampling data from the WC model (Figure 8). In this case, we find that cheaper Flash data (e.g., 0.15 cheaper) can still outperform SE data from Gemini-Pro.
> > > - Finally, we show that price-matched sampling from Gemini-Flash can benefit over Pro for Chat data too (Appendix B.2).
> > >
> > > Q: Key implementation details relegated to appendices
> > > - We thank the review for the suggestion. We will bring the key implementation details in the main text in the revised paper.
> > >
> > > [1] V-STaR: Training Verifiers for Self-Taught Reasoners: https://arxiv.org/abs/2402.06457 \
> > > [2] STaR: https://arxiv.org/abs/2203.14465 \
> > > [3] RestEM: https://arxiv.org/abs/2312.06585 \
> > > [4] Minerva: https://arxiv.org/pdf/2206.14858 \
> > > [5] GSM prompt: https://github.com/EleutherAI/lm-evaluation-harness/blob/main/lm_eval/tasks/gsm8k/gsm8k-cot-llama.yaml#L8-L43 \
> > > [6] IRPO: https://arxiv.org/abs/2404.19733 \
> > > [7] LLM as a judge: https://arxiv.org/abs/2306.05685

---

> > > > ### Author Response · Authors · 2024-11-23
> > > > **Rebuttal reminder**
> > > >
> > > > Hi,
> > > >
> > > > Thanks again for your insightful feedback on our work! We've carefully worked to address your comments/questions. Are there any further questions or concerns we should discuss?

---

> ### Comment · Reviewer_6hvc · 2024-11-26
>
> I appreciate your comprehensive responses to my concerns and questions. You have addressed the key points thoroughly and convincingly.
>
> **Theoretical Understanding & Framework**
> + You've clarified that while formal theoretical bounds weren't the paper's focus, you have meaningful empirical insights about the relationship between coverage, diversity, and FPR
> + The suggested experimental design for understanding WC vs SE performance through feature analysis is promising
> + Your explanation of compute-matched sampling ratios is clear and well-justified
>
> **Methodology & Evaluation**
> + The appendix results on different filtering strategies (including no filtering and LM-as-judge) significantly address my concerns about reliance on ground truth
> + Your justification for the human evaluation sample size makes more sense given the correlation with large-scale automatic evaluation
> + The prompting details and temperature choices are well-explained and consistent with prior work
>
> **Generalisation & Scope**
> + The results on instruction-following tasks demonstrate broader applicability than initially apparent
> + Your work with MATH dataset (including level-specific analysis) and Functional MATH shows good coverage of complex reasoning
> + The exploration across multiple model sizes (7B to 27B) and both open/closed models provides good coverage
>
> **Implementation & Practicality**
> + The cost analysis showing 0.15x cheaper Flash data can outperform Pro is particularly compelling
> + The straightforward integration with existing training pipelines is encouraging
> + The extension to different domains (math, code, instruction following) with minimal modifications strengthens the paper's practical value
>
> *Score Update*: Given the thoroughness of your responses and the additional context provided, particularly around:
> 1. The broader applicability beyond just mathematical reasoning
> 2. The existence of results without ground truth filtering
> 3. The compelling cost-effectiveness analysis
> 4. The clear empirical insights into when WC sampling works better
>
> I am raising my score from 6 to 8. The paper makes a stronger contribution than I initially assessed, with practical implications for making model training more efficient and accessible.
>
> One suggestion for camera-ready: Consider moving some key implementation details and the non-ground-truth filtering results from the appendix to the main paper, as these strengthen your core arguments.

---

> > ### Author Response · Authors · 2024-11-26
> > **Response to the reviewer**
> >
> > Hi,
> >
> > We thank the reviewer for increasing their score. We clarify that the revised paper already has the key implementation details and non-ground-truth filtering results in the main text.

---

### Author Response · Authors · 2024-11-24
**Revised paper update**

We have uploaded the revised version of the paper which addresses most of the comments from the reviewers (highlighted in blue).

---

### Meta-Review · Area_Chair_v4CD · 2024-12-20

**Metareview:**

The authors study scaling for post-training data and whether it's cost (either in $ or compute) efficient to use more data from weaker models vs less data from stronger ones. The authors show in extensive experiments that weaker models can give gains over stronger models.

The paper studies a timely and important area (synthetic data and scaling) and has an interesting take on the problem (weak models are better than strong models). As a drawback, however, the paper's focus on post-training synthetic data limits it given the comparatively low cost of this type of approach compared to human-generated post-training data or pre-training scaling. As reviewer gC6v notes, "I see the paper as a scientifically thorough study of a niche problem." Reviewer rLZh's comment "While the paper aims to highlight the lower computational cost of the WC model for data synthesis (particularly important for large-scale data generation), all the experiments are conducted on relatively small datasets. This discrepancy undermines the overall contribution of the paper." this comes from a similar place of arguing that the generally low overall cost of these settings limits is broader applicability.

**Additional Comments On Reviewer Discussion:**

Reviewer gC6v and the authors had a productive rebuttal discussion, where many of gC6v's concerns were resolved, with the exception of the broad applicability of the approach.

---

### Decision · Program_Chairs · 2025-01-22

Accept (Poster)